# Symmetrization of Loss Functions for Robust Training of Neural Networks in the Presence of Noisy Labels

## Abstract

Labeling a training set is not only often expensive but also susceptible to errors. Consequently, the development of robust loss functions to handle label noise has emerged as a problem of great importance. The symmetry condition provides theoretical guarantees for robustness to such noise. In this work, we investigate a symmetrization method that arises from the unique decomposition of any multi-class loss function into a sum of a symmetric loss function and a class-insensitive term. Notably, the special case of symmetrizing the cross-entropy loss leads to a multi-class extension of the unhinged loss function. This loss function is linear, but unlike in the binary case, it must have specific coefficients in order to satisfy the symmetry condition. Under appropriate assumptions, we demonstrate that this multi-class unhinged loss function is the unique convex multi-class symmetric loss function. It holds a significant role among multi-class symmetric loss functions since the linear approximation of any symmetric loss function around points with equal components must be equivalent to the multi-class unhinged. Furthermore, we introduce SGCE and $\alpha$-MAE, two novel loss functions that smoothly transition between the multi-class unhinged loss and the Mean Absolute Error (MAE). Our experiments demonstrate competitive performance compared to previous state-of-the-art robust loss functions on standard benchmarks, highlighting the effectiveness of our approach in handling label noise.

## 1 Introduction

In recent years, deep learning has made significant advancements, achieving state-of-the-art performance in various domains such as computer vision and natural language processing (LeCun et al., 2015). However, these deep learning models often require extensive training on large datasets. Acquiring correct labels for such datasets can be costly. To mitigate this problem, crowdsourcing platforms have been employed, but they come with the drawback of potentially introducing high amount of errors into the labels. Zhang et al. (2017) tried to fit random labels on CIFAR10 and ImageNet with different deep neural network architectures. They came to the conclusion that deep networks can easily fit random labels during training and that their effective capacity is sufficient for memorizing the entire data set. This memorization ability can become particularly problematic in the presence of label noise, as the network may learn to fit the noise rather than the true underlying patterns. In order to address this problem, robust loss functions based on the symmetry condition have been proposed (Ghosh et al., 2015), Ghosh et al. (2017). A loss function $L(z, y)$, where $z$ is a score vector for the $C$ classes and $y$ is the label, is symmetric if the quantity $\sum_{k=1}^{C} L(z, k)$ remains constant regardless of $z$. Additional explanations are provided in Appendix A. Under the symmetry condition, the optimizers of the expected loss on the clean distribution are the same as the optimizers of the expected loss on the corrupted distribution with uniform label noise. Noise robustness results for some types of non uniform noise can also be obtained (Ghosh et al., 2015), Ghosh et al. (2017). The study and the design of new symmetric loss functions is therefore of great practical importance.

This work proposes a principled symmetrization method for multi-class loss functions leading to a general method for producing symmetric loss functions from non-symmetric loss functions. Our main contributions are the following:

i) We apply the proposed symmetrization method to different loss functions (section 4). Notably, the symmetrization of the cross-entropy loss (CE) leads to a multi-class extension to the unhinged loss function (van Rooyen et al., 2015), (Zhou et al., 2023). Moreover, the symmetrization of the generalized cross-entropy (GCE) (Zhang and Sabuncu, 2018) gives rise to a loss function that smoothly transitions between the multi-class unhinged loss and the mean absolute error (MAE). We refer to this loss as SGCE.

ii) We show the following results about the multi-class unhinged loss function (section 5):

a) It is the unique convex, non-trivial, non-increasing, multi-class symmetric loss function under the assumption of invariance to permutations (defined in section 3).

b) It is the linear approximation to the cross-entropy loss function and to any symmetric loss functions satisfying invariance to permutations at points with equal components (section 5.2).

iii) We introduce $\alpha$-MAE a loss function combining the unhinged loss with the MAE where the parameter $\alpha$ directly controls the $\beta$-smoothness of the loss (section 6).

Experiments comparing SGCE and $\alpha$-MAE with previously proposed robust loss functions on symmetric, asymmetric and natural noise show promising results for our approach (section 7). The proofs of all theoretical results are provided in Appendix D.

## 2 RELATED WORK

Natarajan et al. (2013) proposed a loss correction approach in the binary case which was extended to the multi-class case in Patrini et al. (2016b) by estimating a noise transition matrix. In order to facilitate the estimation of this transition matrix, Yao et al. (2020) considered a factorization of the matrix in two easier to estimate matrices. For its part, Li et al. (2021) estimated the transition matrix and learned the classifier simultaneously (end-to-end). In the terminology of Algan and Ulusoy (2021), the different approaches above belong to the family of noise model based methods since they try to estimate the noise structure directly. On the other hand, noise model free methods mainly try to design intrinsically robust loss functions or to exploit different forms of regularization. Our work is concerned with the problem of designing new robust loss functions (through a process of symmetrization of a loss function) and so it belongs to the family of noise model free methods. An advantage of such methods is that they can be computationally less costly since they do not require to estimate the noise model.

Ghosh et al. (2015) introduced the symmetry condition in the binary case and showed that it is a sufficient condition to make risk minimization robust to label noise. The sigmoid loss, the ramp loss and the probit loss satisfy this condition but the common convex loss functions do not. It turns out that the only binary convex loss function to be symmetric is the unhinged loss (van Rooyen et al., 2015). While the sigmoid loss, the ramp loss and the probit loss achieve robustness by reducing the impact of wrongly classified examples during training (more likely to be corrupted), the unhinged loss achieves robustness by increasing the impact of correctly classified examples during training (more likely to be clean) by being negatively unbounded.

Ghosh et al. (2017) proved label noise robustness results for multi-class symmetric loss functions. Under the symmetry condition, any minimizer of the risk on the corrupted distribution with uniform label noise is also a minimizer of the risk on the clean distribution. Noise tolerance under simple non uniform noise and class conditional noise are also obtained but under some more assumptions (the true risk for the optimal classifier must be 0). They propose the Mean absolute error (MAE) as a robust loss function for training neural networks.

Their experimental results show that while the cross-entropy loss eventually severely fits the noise, the MAE is much more robust. However, the test performance on clean data is often better for the cross-entropy loss. The MAE loss is seen to be more prone to underfitting and to be slower to train because the gradient can saturate while training.

In order to exploit both the noise robustness of MAE and the speed of training of CE, Zhang and Sabuncu (2018) proposes a generalization of the two losses. Their loss function is the negative Box-Cox transformation and it allows to interpolate between the MAE and the CE with a hyperparameter. The symmetric cross-entropy loss is proposed in Wang et al. (2019). This loss function is composed of two terms. A robust term (reverse cross-entropy loss) and the standard cross-entropy loss (for convergence). The reverse cross-entropy loss is a scalar multiple of the MAE. Taylor cross entropy loss (Feng et al., 2021) considers approximations to the cross-entropy loss of different orders. The MAE is the first order Taylor approximation of CE. The second-order Taylor Series approximation of CE is an average combination of MAE and a lower bound of Mean Squared Error (MSE). Order 2 and above approximations of the CE are not symmetric loss functions. Considering different orders of approximation to the CE is another way to interpolate between the MAE and the CE. These different methods are therefore all trying to make a compromise between the robustness of the MAE and the better fitting ability of CE. Our method is always guaranteed to lead to a symmetric loss function (contrary to the approaches above) and the problem of underfitting can be alleviated by controlling the amount of saturation in the loss.

In the special case of the cross-entropy loss, our symmetrization method leads to a multi-class extension of the unhinged loss function. In most previous works, the unhinged loss in the multi-class setting was taken to be equivalent to the MAE (Zhang and Sabuncu, 2018), (Zhu et al., 2023). This is not the case in our work. In our case, the multi-class unhinged loss is a linear symmetric loss function like its binary counterpart. This means that no softmax function is being used at the final layer. The work of Patrini et al. (2016b) considers a loss function that they call unhinged in their experiments, but it is actually not a symmetric loss function. It is equivalent up to an additive constant and a multiplicative constant to minus the logit at the target label. The same multi-class extension of the unhinged loss as ours is explored in (Zhou et al., 2023), but in a different context and for a different purpose. They obtain the multi-class unhinged loss by removing the maximum in the multi-class hinge loss. Their goal is to simplify the theoretical analysis of gradient descent dynamics, which they argue is harder to investigate using cross-entropy or mean squared error. Being linear, the unhinged loss can simplify theoretical research. Our focus, however, is on discussing the fundamental role of the multi-class unhinged loss among multi-class symmetric loss functions and proposing new robust loss functions (SGCE, $\alpha$-MAE) that build upon the unhinged loss.

The closest work to our own is (Ma et al., 2020). They propose a normalization applicable to any loss function that leads to a symmetric loss function. They observe however that their robust loss functions often suffer from underfitting. The solution that they propose is called the active-passive loss framework. An active loss function is only optimizing directly at the specified label. A passive loss also explicitly minimizes the probability for other classes. An active-passive loss is then defined to be a weighted combination of an active and a passive loss. Both the active and the passive loss are required to be robust in order to ensure that the combination is also robust. Subsequently, Ye et al. (2023) proposed the normalized negative cross-entropy as a substitute to the MAE in the active-passive framework. Our work instead proposes a different general method than normalization to produce symmetric loss functions. Our method exploit the fact that there is a symmetric loss function hidden within any loss function, as demonstrated by our general decomposition result.

The work of Patrini et al. (2016a) investigates binary loss functions that can be decomposed as a sum of a class-insensitive term and a linear term. They refer to these losses as linear-odd losses (the odd part of the loss is linear). The labeled data centroid becomes the quantity of interest in the linear component of the loss and the subsequent works of Gao et al. (2016), Gong et al. (2021) and Gong et al. (2022) proposed estimators for this centroid when only corrupted data is available. Ding et al. (2022) extended these ideas to the multi-class case by decomposing the multi-class mean squared error loss in order to also reduce the problem to centroid estimation. Our work considers instead the general decomposition of any multi-class

loss function into a symmetric loss function and a class-insensitive term. We discuss the multi-class data centroid related to the multi-class unhinged loss function in Appendix C.

## 3 Assumptions on multi-class loss functions

We consider loss functions of the form $L(z, y)$, where $z = (z_i, \cdots, z_C) = h(x) \in \mathbb{R}^C$ is the score vector for some neural network $h$, $x \in \mathbb{R}^d$ is an input and $y \in \{1, \cdots, C\}$ is a label for an example $(x, y)$ sampled from a distribution $D$ on $\mathbb{R}^d \times \{1, \cdots, C\}$. We will make two main assumptions on the loss function. First, we want the loss function $L(z, y)$ to be non-increasing.

**Definition 3.1** *We say that $L(z, y)$ is a non-increasing multi-class loss function if for all $y$, the function $L(z, y)$ is a non-increasing function of $z_y$ when $z_k$ for $k \neq y$ is kept fixed.*

Secondly, we want a form of symmetry between the classes to hold (different from the notion of symmetry related to noise robustness). This is done to ensure that $L(z, k)$ and $L(z', k')$, where $k \neq k'$, will be the same function when the roles of $k$ and $k'$ are swapped. We will refer to this property as *invariance to permutations* and it is defined precisely below.

**Definition 3.2** *The loss function $L(z, y)$ is invariant to permutations $\tau$ on $C$ elements if $L(\tau(z), \tau(y)) = L(z, y)$ for all $z$, $y$, and $\tau$. A permutation $\tau$ acts on $z$ by permuting the components of $z$, that is, $\tau(z) = (z_{\tau^{-1}(1)}, \cdots, z_{\tau^{-1}(C)})$.*

An example of a loss function satisfying both assumptions above is the standard cross-entropy loss defined by $L(z, y) = -\log(p_y)$, where $p_y = \frac{\exp(z_y)}{\sum_{k=1}^C \exp(z_k)}$ is obtained via the softmax function. Another simple example is the linear loss function $L(z, y) = -z_y$. An example of a loss function that does not satisfy the invariance to permutations property is $L(z, y) = -y z_y$.

## 4 Symmetrization of loss functions

We ask the question of how to decompose a loss function into a sum of a symmetric loss function and a class-insensitive term. It happens to be the case that there is a unique such decomposition up to constants.

**Proposition 4.1** *There is a unique (up to constants) decomposition of a loss function into a sum of a symmetric loss function and a class-insensitive term. The symmetric component is given by*

$$L^{sym}(z, y) := L(z, y) - \frac{1}{C} \sum_{k=1}^C L(z, k). \tag{1}$$

It is not difficult to verify that if $L(z, y)$ satisfies the property of invariance to permutations, then $L^{sym}(z, y)$ must also satisfy it. In the following subsections, we apply equation 1 to different loss functions.

### 4.1 Symmetrization of the cross-entropy loss

It is easy to verify that the symmetrization of the multiclass cross-entropy loss is the linear function

$$-z_y + \frac{1}{C} \sum_{k=1}^C z_k. \tag{2}$$

We refer to 2 as the *multi-class unhinged loss* (see also (Zhou et al., 2023)). We note that since $-\log(p_y) = -z_y + \log(\sum_{k=1}^C \exp(z_k))$ and the term $\log(\sum_{k=1}^C \exp(z_k))$ is class-insensitive, the symmetrization of the cross-entropy loss is the same as the symmetrization of the linear loss $-z_y$.

In cases where the original loss function is the negative log-likelihood, our symmetrization method can be interpreted as a form of regularization, induced by applying data-dependent Dirichlet priors to the network's outputs in a specific amount. Further details on this interpretation are provided in Appendix E.

## 4.2 Symmetrization of the mean squared error

The mean squared error for classification is given by

$$L(z, y) = ||e_y - s(z)||_2^2 = ||s(z)||_2^2 + 1 - 2p(y|z),$$

where $e_y$ is the one-hot encoding for the label $y$, $s$ is the softmax function and the $y^{th}$ coordinate of $s(z)$ is $p(y|z)$. Since the term $||s(z)||_2^2$ is independent of the label, the symmetrization operator removes it and we are left with a loss equivalent to the MAE (the MAE is defined as $1 - p(y|z)$).

## 4.3 Symmetrization of the generalized cross-entropy loss (SGCE)

The generalized cross-entropy loss (GCE) (Zhang and Sabuncu, 2018) is defined by

$$L_q(z, y) := \frac{1 - p(y|z)^q}{q},$$

where $q \in (0, 1]$. When $q$ goes to 0, the loss converges to the cross-entropy. When $q = 1$, we get the MAE. The symmetrization of the generalized cross-entropy loss (SGCE) leads therefore to a form of interpolation between the multi-class unhinged and the MAE. The unhinged loss function is robust by maintaining larger gradients for examples already correctly classified. The MAE is robust by reducing the gradient of incorrectly classified examples (since they might be corrupted). The SGCE loss function allows to realize a trade-off between these two strategies.

## 4.4 Symmetrization of the cosine similarity loss

Consider the cosine similarity loss between the score vector $z$ and the one-hot encoding $e_y$ for the label: $1 - \frac{z \cdot e_y}{||z|| ||e_y||}$. The symmetrization of this loss is the multi-class unhinged but with the normalization of $z$ instead of $z$ as input. This is a simple way to address the fact that the unhinged is negatively unbounded (see 7.2 for experimental details).

## 5 Some properties of the multi-class unhinged

### 5.1 Uniqueness among convex and symmetric loss functions

The multi-class unhinged loss satisfies the same fundamental result as the binary unhinged loss.

**Theorem 5.1** *The multi-class unhinged loss is the unique convex, non-trivial, non-increasing, multi-class symmetric loss function satisfying the property of invariance to permutations (up to an additive and a multiplicative constant).*

The invariance to permutations property plays a crucial role in the proof for the multi-class case. The initial step in our proof involves establishing constraints on the coefficients of a linear loss that satisfies the invariance to permutations property (Lemma D.1). These constraints on the coefficients are subsequently used to prove uniqueness among linear loss functions (Proposition D.2). The proof can then be completed by observing that any convex symmetric loss function must be both convex and concave, and therefore is affine.

### 5.2 Linear approximations of multi-class loss functions

Since not any linear loss function is symmetric in the multi-class setting, it is an interesting question to ask when the linear approximation around a point $z'$ (for example the origin) is

symmetric. If this is the case, we can expect the loss function to be robust when training stays around $z'$. Regularization methods like weight decay, batch normalization and early stopping can then all contribute to maintain training in the robust region.

It happens to be the case that the linear approximation of the cross-entropy loss around $z = z'$, where the components of $z'$ are all equal, is equivalent to the multi-class unhinged loss function. Indeed, the gradient of the cross-entropy loss with respect to $z$ is given by

$$\nabla_z L(z, y) = s(z) - e_y,$$

where $s(z)$ is the output of the softmax function evaluated at $z$. The linear approximation of $L(z, y)$ at $z'$ is then given by

$$(s(z') - e_y)^T (z - z') + L(z', y),$$

where $T$ denotes transposition. Up to constants from the point of view of the variable $z$, we only need to consider $(s(z') - e_y)^T z$. Since $s(z') = (\frac{1}{C}, \cdots, \frac{1}{C})$ when all the components of $z'$ are equal,

$$(s(z') - e_y)^T z = (\frac{1}{C} - 1)z_y + \frac{1}{C} \sum_{k \neq y} z_k = -z_y + \frac{1}{C} \sum_{k=1}^{C} z_k.$$

We conclude that the linear approximation of the cross-entropy loss around $z'$ is equivalent to the multi-class unhinged loss function. The cross-entropy loss is therefore "locally symmetric" around any such $z'$. This can help to explain why the cross-entropy loss can already be somewhat robust in particular when early stopping is being used. Indeed, if training stays close enough to an initial point such that the probability for each class is the same, we are approximately training with a symmetric loss function.

We found an example of a non-robust loss function that happens to be locally equivalent to the multi-class unhinged loss function at some specific points. If the loss function is globally robust, it is actually guaranteed to be locally equivalent to the multi-class unhinged loss function at every point $z'$ with equal components (that is not a critical point).

**Proposition 5.2** *Assume that $L(z, y)$ is non-increasing, symmetric, satisfies the property of invariance to permutations and is differentiable at $z'$ a vector with equal components. Then, the linear approximation of $L(z, y)$ at $z'$ is equivalent to the multi-class unhinged loss function if $\nabla_z L(z, y)|_{z=z'} \neq 0$.*

$\beta$-smoothness allows to bound the size of the remainder of the linear approximation to a loss function $\phi(z, y)$. We can then give a quantitative result about the gap between the solution obtained with $\phi$ and the multi-class unhinged solution. A smaller $\beta$ and more regularization will lead to a solution with a closer unhinged risk to the optimal unhinged risk when optimizing with the loss $\phi$.

**Proposition 5.3** *Consider an euclidean ball of radius $R$ around $0 \in \mathbb{R}^d$ for the domain of $x$. Assume that the loss $\phi(z, y)$ is $\beta$-smooth for all $y$ and that its linear approximation at $0$ is the multi-class unhinged loss function (denoted by $L(z, y)$). Furthermore, assume that $l$ is the number of layers of the feedforward neural network $f$, the non-linearity is $c$-Lipschitz at every layer and the product of the euclidean norm of the weights of each layer is bounded by $r$. Then,*

$$L_D(f_\phi^*) - L_D(f_L^*) \leq R^2 \beta c^{2(l-1)} r^2,$$

*where $f_\phi^*$ is a minimizer for the true risk for loss $\phi$ and $f_L^*$ is a minimizer for the true risk for the multi-class unhinged loss. This means that $f_\phi^*$ will reach a similar unhinged risk to the optimal unhinged risk (if $\beta$ and $r$ are small).*

## 6    Combining the Multi-Class Unhinged with Non-Linear Multi-Class Symmetric Loss Functions

From Proposition 5.2, any symmetric loss function $L(z, y)$ satisfying the assumptions of the proposition can be expressed as:

$$L(z, y) = L(0, y) + \text{Constant} \left( -z_y + \frac{1}{C} \sum_{k=1}^{C} z_k \right) + g(z, y),$$

where $g(z, y)$ represents the residual of the linear approximation of $L(z, y)$ at $z = 0$. A straightforward way to control the degree of non-linearity in the loss function is to introduce a hyperparameter $\alpha \in [0, \infty)$ in front of $g(z, y)$. Let this new loss function be denoted as $L_\alpha(z, y)$. If $L(z, y)$ is $\beta$-smooth, then $L_\alpha(z, y)$ is $\alpha\beta$-smooth. Hence, $\alpha$ controls the $\beta$-smoothness of the loss.

Without loss of generality, assume that Constant $= 1$ (otherwise, rescale the loss accordingly). Then, up to an additive constant, the loss function can be written as:

$$L_\alpha(z, y) = (1 - \alpha)L_0(z, y) + \alpha L(z, y),$$

where $L_0(z, y)$ is exactly equal to the multi-class unhinged loss function. We apply this approach to the MAE loss and refer to the resulting loss as $\alpha$-MAE. In this case, the constant in front of the unhinged loss is $\frac{1}{C}$, so we rescale the MAE by $C$, leading to the following expression:

$$\alpha\text{-MAE} = (1 - \alpha) \left( -z_y + \frac{1}{C} \sum_{k=1}^{C} z_k \right) + \alpha C \left( 1 - p(y|z) \right),$$

for $\alpha \in [0, \infty)$.

## 7    Experiments

### 7.1    Method

We first compare the performance of the multi-class unhinged loss, SGCE, and $\alpha$-MAE against various robust loss functions on CIFAR-10 and CIFAR-100 (Krizhevsky, 2009), as presented in Table 1. For the CIFAR-10 experiments, we trained an 8-layer CNN for 120 epochs, while for CIFAR-100, we used a ResNet-34 (He et al., 2016) architecture and trained it for 200 epochs. The comparison includes CE, MAE, GCE (Zhang and Sabuncu, 2018), SCE (Wang et al., 2019), NCE+RCE (Ma et al., 2020), NCE+AGCE (Zhou et al., 2021), and ANL-CE (Ye et al., 2023). Both symmetric label noise (see A) and asymmetric label noise (non-uniform corruption probabilities based on class similarities, as described in (Patrini et al., 2016b)) were considered.

To ensure a fair comparison, we implemented our method within the public implementation of (Ye et al., 2023). The key difference is that we tuned the weight decay term separately for each loss function, whereas (Ye et al., 2023) only tuned an additional regularization parameter $\delta$ for their method without adjusting the weight decay for other loss functions. Otherwise, we followed the same experimental protocol as in (Ma et al., 2020) and (Ye et al., 2023). On CIFAR-10, the weight decay was tuned over the set $\{1 \times 10^{-4}, 5 \times 10^{-4}, 1 \times 10^{-3}, 5 \times 10^{-3}, 1 \times 10^{-2}\}$, and for CIFAR-100, over $\{1 \times 10^{-5}, 5 \times 10^{-5}, 1 \times 10^{-4}, 5 \times 10^{-4}, 1 \times 10^{-3}\}$. The hyperparameter $q$ for SGCE was selected from $\{0.2, 0.35, 0.50, 0.65, 0.80\}$, and the hyperparameter $\alpha$ for $\alpha$-MAE was chosen from $\{0.25, 0.50, 1.0, 2.0, 4.0\}$.

Hyperparameters were tuned using 10% of the training data as a validation set, based on a symmetric noise rate of 80%. These hyperparameters were then used across all other noise rates, including asymmetric noise. The training algorithm used in all cases was SGD with momentum. The learning rate and other SGD parameters were not tuned. For example, the learning rate for CIFAR-10 was fixed at 0.01 and for CIFAR-100 at 0.1, consistent with (Ye et al., 2023). Furthermore, we used a cosine annealing schedule to maintain consistency and ensure fair comparison with (Ma et al., 2020) and (Ye et al., 2023). While this learning rate

Table 1: Accuracy (mean of 3 runs with standard deviation in parentheses) of the multi-class unhinged, SGCE and $\alpha$-MAE compared to previously proposed robust loss functions on CIFAR10 and CIFAR100 with symmetric noise rate $\eta \in \{0.4, 0.6, 0.8\}$ and asymmetric noise rate $\eta \in \{0.2, 0.3, 0.4\}$. The best result for each case is in bold.

| Datasets | Loss functions | Clean | Symmetric Noise Rate ($\eta$) | | | Asymmetric Noise Rate ($\eta$) | | |
|---|---|---|---|---|---|---|---|---|
| | | | 0.4 | 0.6 | 0.8 | 0.2 | 0.3 | 0.4 |
| CIFAR10 | CE | 93.45(0.30) | 69.69(0.52) | 51.88(0.37) | 32.59(0.76) | 85.84(0.26) | 81.08(0.47) | 75.43(0.21) |
| | MAE | 88.80(0.17) | 84.33(0.12) | 77.27(0.24) | 47.86(0.48) | 84.47(2.65) | 66.56(4.76) | 58.72(2.24) |
| | GCE | 93.48(0.04) | 74.28(0.13) | 56.30(0.44) | 39.88(2.11) | 85.96(0.17) | 80.78(0.37) | 75.34(0.30) |
| | SCE | 92.99(0.06) | 87.85(0.36) | 79.80(0.16) | 22.43(2.18) | 90.10(0.06) | 85.29(0.41) | 76.26(0.15) |
| | NCE+RCE | 90.94(0.01) | 86.03(0.13) | 79.89(0.11) | 55.52(2.74) | 88.36(0.13) | 84.84(0.16) | 77.75(0.37) |
| | NCE+AGCE | 91.08(0.06) | 86.16(0.10) | 80.14(0.27) | 55.62(4.78) | 88.48(0.09) | 84.79(0.15) | **78.60(0.41)** |
| | ANL-CE | 91.66(0.04) | 87.28(0.02) | 81.12(0.30) | 61.27(0.55) | 89.13(0.11) | 85.52(0.24) | 77.63(0.31) |
| | Unhinged | 93.03(0.11) | 86.21(0.31) | 76.58(0.20) | 50.47(0.89) | 88.48(0.33) | 83.11(0.15) | 76.55(0.23) |
| | SGCE | 93.05(0.22) | 87.58(0.12) | 79.57(0.48) | 61.06(1.60) | 89.38(0.07) | 83.20(0.45) | 75.39(0.39) |
| | $\alpha$-MAE | 92.64(0.18) | **88.17(0.15)** | **81.82(0.62)** | **62.08(1.24)** | **90.07(0.31)** | **86.11(0.17)** | 77.02(0.65) |
| CIFAR100 | CE | 77.25(0.60) | 47.75(0.31) | 29.03(0.23) | 14.74(0.44) | 63.70(0.23) | 55.35(0.54) | **45.49(0.15)** |
| | MAE | 16.74(2.18) | 7.29(0.89) | 3.78(0.74) | 2.42(0.96) | 7.44(0.75) | 6.30(0.56) | 5.62(0.30) |
| | GCE | 61.37(0.71) | 56.42(0.37) | 46.31(1.01) | 21.46(0.06) | 55.27(1.07) | 48.05(0.81) | 40.20(0.56) |
| | SCE | 76.37(0.19) | 48.44(0.14) | 30.58(0.57) | 11.65(1.38) | 63.02(0.01) | 54.52(0.28) | 44.87(0.50) |
| | NCE+RCE | 68.22(0.28) | 57.97(0.30) | 46.26(1.07) | 25.65(0.51) | 62.77(0.53) | 55.62(0.56) | 42.46(0.42) |
| | NCE+AGCE | 68.61(0.22) | 59.74(0.68) | 47.96(0.44) | 24.13(0.07) | 64.05(0.25) | 56.36(0.59) | 44.90(0.62) |
| | ANL-CE | 70.68(0.23) | 61.80(0.50) | 51.52(0.53) | 28.07(0.28) | 66.27(0.19) | **59.76(0.34)** | 45.41(0.68) |
| | Unhinged | 74.65(0.44) | 59.90(0.03) | 44.54(0.35) | 20.97(0.67) | 59.14(0.21) | 50.16(0.30) | 42.56(0.28) |
| | SGCE | 74.62(0.15) | 63.48(0.30) | 50.25(0.51) | **31.56(0.42)** | 60.39(0.36) | 49.08(0.36) | 41.05(0.23) |
| | $\alpha$-MAE | 73.96(0.26) | **65.90(0.23)** | **56.42(0.19)** | 29.89(0.64) | **68.30(0.52)** | 56.01(0.45) | 42.04(0.31) |

schedule may not optimize performance, as shown in Table 4—where we report results on CIFAR-100 using a constant learning rate divided by 10 at 95% of training (one-step decay of learning rates)—it is a commonly used schedule and was kept to ensure consistency with previous work. More details about the training configuration and hyperparameters are given in Appendix G.

We also evaluated our approach in settings with natural noise, comparing it to previously proposed robust loss functions. Results for CIFAR-10N and CIFAR-100N (Wei et al., 2022) are shown in Table 2. The same hyperparameters used for CIFAR-10 and CIFAR-100 were applied in these experiments without further tuning. Additionally, we report SGCE results on Mini WebVision (using the Google-resized data and the first 50 classes) (Li et al., 2017), (Jiang et al., 2017). Performance was evaluated on both the WebVision validation set and the ILSVRC 2012 validation set (Russakovsky et al., 2015), with results presented in Table 3.

### 7.2 Normalization of the score vector

Since the loss functions resulting from the symmetrization operation and involving the multi-class unhinged loss can be negatively unbounded, a method to prevent numerical overflows is required. We considered two approaches: applying Euclidean normalization to the score vector and adding a batch normalization layer to the score vector. To ensure numerical stability in Euclidean normalization, an epsilon value of $1 \times 10^{-5}$ is used to prevent division by zero by clamping the denominator away from 0. Using batch normalization on the score vector was also considered in (Patrini et al., 2016b).

When using the cosine annealing schedule for learning rates, Euclidean normalization consistently outperformed batch normalization. The results in Tables 1 and 2 were obtained with Euclidean normalization applied to the unhinged loss, SGCE, and $\alpha$-MAE. Under the one-step learning rate decay schedule, SGCE performed better with batch normalization, while both the unhinged loss and $\alpha$-MAE achieved better results with Euclidean normalization. Results obtained using batch normalization are indicated by "(BN)", while all other results reported were obtained with Euclidean normalization.

## 8 Discussion of results

Overall, SGCE and $\alpha$-MAE maintain competitive performance across varying noise levels and datasets, showing that the combination of the multi-class unhinged loss with MAE leads to robustness in both synthetic and natural noise scenarios. $\alpha$-MAE particularly shines on

Table 2: Accuracy (mean of 3 runs with standard deviation in parentheses) of the multi-class unhinged, SGCE and $\alpha$-MAE compared to previously proposed robust loss functions on CIFAR-10N and CIFAR-100N. The best result for each case is in bold.

| Loss functions | CIFAR-10N | | | | | CIFAR-100N |
| --- | --- | --- | --- | --- | --- | --- |
| | Aggregate | Random 1 | Random 2 | Random3 | Worst | |
| NCE+RCE | 89.17(0.28) | 87.62(0.34) | 87.66(0.12) | 87.70(0.18) | 79.74(0.09) | 54.27(0.09) |
| NCE+AGCE | 89.27(0.28) | 87.92(0.02) | 87.61(0.20) | 87.62(0.16) | 79.91(0.37) | 55.96(0.20) |
| ANL-CE | 89.66(0.12) | 88.68(0.13) | 88.19(0.08) | 88.24(0.15) | 80.23(0.28) | 56.37(0.42) |
| Unhinged | 90.34(0.26) | 88.53(0.13) | 88.10(0.21) | 88.41(0.19) | 77.24(0.19) | 54.33(0.34) |
| SGCE | 90.62(0.18) | 89.19(0.02) | 88.92(0.14) | 88.94(0.23) | 78.47(0.35) | 56.31(0.39) |
| $\alpha$-MAE | **90.67(0.17)** | **89.57(0.13)** | **89.37(0.03)** | **89.49(0.23)** | **81.28(0.37)** | **59.41(0.31)** |

Table 3: Accuracy of SGCE compared to previously proposed robust loss functions when training a ResNet-50 on the WebVision training data set. Performance on the ILSVRC 2012 validation data and the WebVision validation data are reported. The best result is in bold for each case.

| Methods | CE | GCE | SCE | NCE+RCE | NCE+AGCE | ANL-CE | ANL-FL | SGCE(BN) |
| --- | --- | --- | --- | --- | --- | --- | --- | --- |
| ILSVRC12 Val | 58.64 | 56.56 | 62.60 | 62.40 | 60.76 | 65.00 | 65.56 | **69.52** |
| WebVision Val | 61.20 | 59.44 | 68.00 | 64.92 | 63.92 | 67.44 | 68.32 | **74.04** |

Table 4: Accuracy (mean of 3 runs with standard deviation in parentheses) of the multi-class unhinged, SGCE and $\alpha$-MAE compared to previously proposed robust loss functions on CIFAR100 with a one-step decay of learning rates. The best result for each case is in bold.

| Loss functions | Clean | Symmetric Noise Rate ($\eta$) | | | Asymmetric Noise Rate ($\eta$) | | |
| --- | --- | --- | --- | --- | --- | --- | --- |
| | | 0.4 | 0.6 | 0.8 | 0.2 | 0.3 | 0.4 |
| CE | 75.15(0.26) | 50.57(0.32) | 35.64(0.03) | 21.70(0.41) | 66.86(0.38) | 59.75(1.05) | 49.08(0.41) |
| GCE | 66.55(0.78) | 64.09(0.90) | 58.41(0.48) | 38.46(0.63) | 60.96(0.67) | 57.59(0.87) | 48.95(1.27) |
| ANL-CE | 73.11(0.15) | 65.09(1.43) | 58.62(0.59) | 32.93(0.53) | 69.91(0.24) | 65.06(0.26) | 51.98(0.11) |
| Unhinged | 72.45(0.13) | 65.58(0.38) | 58.29(0.78) | 37.11(0.61) | **70.33(0.25)** | **69.34(0.37)** | 64.84(0.28) |
| SGCE(BN) | 73.43(0.11) | **66.65(0.27)** | **59.74(0.31)** | **43.89(0.91)** | 70.17(0.15) | 63.55(1.70) | 50.88(1.00) |
| $\alpha$-MAE | 71.61(0.24) | 64.35(0.20) | 57.73(0.36) | 37.20(0.99) | 69.24(0.20) | 68.78(0.50) | **65.01(0.50)** |

various datasets and noise rates, for example, consistently outperforming other methods on CIFAR-10N and CIFAR-100N. SGCE performed impressively with the one-step decay of learning rates and symmetric noise. Additionally, both loss functions have only one hyperparameter to tune, whereas methods from the active-passive approach typically use two. Notably, the underfitting issues with MAE on CIFAR-100 are resolved by $\alpha$-MAE.

## 9 CONCLUSION

In this work, we proposed a principled symmetrization method for designing robust loss functions to handle label noise. The symmetrization of the categorical cross-entropy loss leads to the unique convex, non-trivial, non-increasing multi-class symmetric loss function under the technical assumption of invariance to permutations. As such, this loss function extends the binary unhinged loss in the multi-class case. The symmetrization of the generalized cross-entropy loss (SGCE) and the newly introduced $\alpha$-MAE allow for the effective combination of the multi-class unhinged loss with the MAE. Our approach demonstrates competitive performance compared to previously proposed robust loss functions on the benchmark datasets CIFAR-10, CIFAR-100, and WebVision.

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

## A    Uniform label noise and the symmetry condition

The influential work of (Ghosh et al., 2015) and (Ghosh et al., 2017) introduced the concept of symmetric loss functions and established the fundamental results of statistical consistency that they satisfy when training in the presence of noisy labels. At the limit of infinite data, an optimal classifier for the corrupted distribution will also be an optimal solution for the clean distribution if the loss function is symmetric. In this section, we introduce the concept of symmetric loss functions to readers who are new to these ideas.

Assume that with some probability $p$, instead of sampling the label of an example from the true distribution, we sample the label from a uniform distribution on the $C$ classes. In other words, define the corrupted distribution $\overline{D}$ to have the same marginal distribution as $D$ (that is $\overline{D}_x = D_x$) but with conditional distribution $\overline{D}_{y|x}$ given by

$$pU(\{1, \cdots, C\}) + (1 - p)D_{y|x},$$

where $U(\{1, \cdots, C\})$ is a uniform distribution over $\{1, \cdots, C\}$. Given a training set $S$ from $D$, we can corrupt it to get a set $\overline{S}$ by changing the label of an example $(x, y) \in S$ to a different label with probability $\eta = \frac{(C-1)p}{C}$ (each one of the different labels from $y$ having a probability of $\frac{p}{C}$ of being the new label). The probability $\eta$ is usually referred to as the (symmetric) noise rate. We require that $p < 1$ or, equivalently, $\eta < \frac{C-1}{C}$.

It is then straightforward to get

$$L_{\overline{D}}(h) = \frac{p}{C}\left[\mathop{\mathbb{E}}_{x \sim D_x} \sum_{k=1}^{C} L(h(x), k)\right] + (1 - p)L_D(h),$$

where $L_D(h)$ is the expected loss (over distribution $D$) of classifier $h$. If we could get rid of the term $\mathbb{E}_{x \sim D_x} \sum_{k=1}^{C} L(h(x), k)$ above, the true risk on the clean distribution would be proportional to the true risk on the corrupted distribution (if $p < 1$). The optimizers of $L_{\overline{D}}(h)$ would then be the same as the optimizers of $L_D(h)$. This is the motivation for the symmetry condition.

**Definition A.1** *A loss function $L(z, y)$ is said to be symmetric if, for all $z$,*

$$\sum_{k=1}^{C} L(z, k) = constant.$$

## B    Decomposition in the binary case and Taylor series

In the binary case, the unique decomposition of a loss function $\phi(z)$ into a sum of a symmetric loss function and a class-insensitive loss function is the sum of its odd part $\frac{\phi(z)-\phi(-z)}{2}$ with

its even part $\frac{\phi(z)+\phi(-z)}{2}$. If $\phi$ admits a Taylor series expansion around 0, it is possible to characterize the symmetry condition using the coefficients of the Taylor series and to express the odd part and the even part with the series.

**Proposition B.1** *Assume that $\phi$ is an infinitely differentiable potential and without loss of generality that $\phi(0) = 0$. Then, $\phi$ is symmetric if and only if $\phi^{(k)}(0) = 0$ for all $k$ even. That is, $\phi$ is symmetric if and only if the even coefficients of its Taylor expansion at 0 are all 0.*

Since the odd part of $\phi$ is the sum over the terms with odd coefficients of the Taylor series, our symmetrization method corresponds to the very natural process of keeping the odd coefficients of the original loss and replacing the even coefficients by 0's. Every truncation of the Taylor expansion for the symmetric loss function is also symmetric. This allows approximating any such symmetric loss functions with simpler polynomial symmetric loss functions.

The decomposition as a sum of an odd and an even function in the binary case makes sense because changing the label amounts to changing the sign of $z$. However, this does not generalize immediately to the multi-class case. When taking the point of view that the odd binary loss functions are the symmetric loss functions and that the even binary loss functions are the class-insensitve loss functions, we can get a decomposition holding in the multi-class case also.

## C  LINEAR HYPOTHESIS CLASSES AND INTERPRETATION AS KERNEL LEARNING

Assume that we are in the linear multi-class case with feature map $\psi$. Let $W$ be the weight matrix and $L(z, y)$ the multi-class unhinged loss function. If a bias vector $b$ is present, the weight matrix will be understood as being extended by an additional column (the vector $b$) and the vector $\psi(x)$ will be understood as having an additional entry. Consider the constrained optimization problem given by minimizing the empirical multi-class unhinged loss function on the training data under a constraint on the Frobenius norm of the weight matrix ($||W||_{FR} \leq r$). This constraint is equivalent to $||W||_{FR}^2 - r^2 \leq 0$. The Lagrangian is then given by

$$\frac{1}{N} \sum_{i=1}^N L(W\psi(x_i), y_i) + \lambda(||W||_{FR}^2 - r^2),$$

for $\lambda \geq 0$. Define the column vector $c_y$ as having $y^{th}$ entry equal to $\frac{C-1}{C}$ and every other entry given by $\frac{-1}{C}$. The first order condition on the Lagrangian can then be written as

$$2\lambda W - \frac{1}{N} \sum_{i=1}^N c_{y_i} \psi(x_i)^T = 0.$$

Let

$$\mu_S^{unh} := \frac{1}{N} \sum_{i=1}^N c_{y_i} \psi(x_i)^T.$$

We will refer to $\mu_S^{unh}$ as the unhinged multi-class data centroid. The above computations showed that $\nabla_W \left[ \frac{1}{N} \sum_{i=1}^N L(W\psi(x_i), y_i) \right] = -\mu_S^{unh}$, where the gradient is taken with respect to the matrix $W$. The $k^{th}$ row of $-\mu_S^{unh}$ is equal to the gradient of the multi-class unhinged loss with respect to the weight vector connected to the $k^{th}$ output (score for class $k$). If $\mu_S^{unh} = 0$, from the KKT conditions, we get that any $W$ satisfying the constraint is a solution (taking $\lambda = 0$). The problem is degenerate in that case. Indeed, the gradient of the multi-class unhinged loss on the training set is then 0 for any $W$. Assume that $\mu_S^{unh} \neq 0$. Then, from the first order condition on the Lagrangian, we must have $\lambda \neq 0$ and $W = \frac{1}{2\lambda} \mu_S^{unh}$. Furthermore, from the KKT conditions, we must have $||W||_{FR} = r$. Therefore,

$$W = r \frac{\mu_S^{unh}}{||\mu_S^{unh}||_{FR}}.$$

We note that $r$ is only a scaling factor and so the solution is the same for any $r$ from the point of view of classification performance in terms of $0 - 1$ loss. The quantity $\mu_S^{unh}$ does more than offering the solution to our optimization problem, it actually fully characterizes the loss landscape on training data. Indeed, if two training data sets have the same $\mu_S^{unh}$, then their multi-class unhinged loss functions have the same gradient everywhere and so can differ only by a constant. This can also be seen by verifying with a direct computation that

$$\frac{1}{N} \sum_{i=1}^{N} L(W\psi(x_i), y_i) = -Trace(\mu_S^{unh} W^T). \tag{3}$$

Assume now that the feature map is given by a deep neural network with parameters $\theta$. Denote this feature map by $\psi_\theta$ and the unhinged multi-class data centroid by $\mu_{S,\theta}^{unh}$. Furthermore, let $k_\theta(x, x') := \psi_\theta(x)^T \psi_\theta(x')$ be the kernel given by the standard dot product in the representation space. Substituting $W = r \frac{\mu_{S,\theta}^{unh}}{||\mu_{S,\theta}^{unh}||_{FR}}$ in equation 3 allows us to write the empirical loss as a function of $\theta$ only. We denote this empirical loss as $L_S(\theta)$. Since $Trace([\mu_{S,\theta}^{unh}][\mu_{S,\theta}^{unh}]^T) = ||\mu_{S,\theta}^{unh}||_{FR}^2$, we get

$$L_S(\theta) = -r||\mu_{S,\theta}^{unh}||_{FR}.$$

Minimizing $L_S(\theta)$ is therefore an equivalent problem to maximizing $||\mu_{S,\theta}^{unh}||_{FR}^2$.

**Proposition C.1** *The squared Frobenius norm of the unhinged multi-class data centroid satisfies the equality*

$$||\mu_{S,\theta}^{unh}||_{FR}^2 = \frac{1}{N^2} \sum_{i,j} a_{ij} k_\theta(x_i, x_j), \tag{4}$$

*where $a_{ij}$ is equal to $\frac{C-1}{C}$ if $y_i = y_j$ and to $\frac{-1}{C}$ if $y_i \neq y_j$.*

Equation 4 gives a direct and precise interpretation of training neural networks with the multi-class unhinged loss as a form of kernel learning. When $x_i$ and $x_j$ share the same label, the coefficient $a_{ij}$ is positive, and the objective aims to increase the similarity (or alignment) between these points. Conversely, when two points do not share the same label, the coefficient $a_{ij}$ is negative, and the objective seeks to decrease the similarity between the points.

Ding et al. (2022) obtained a multi-class data centroid by decomposing the mean-squared error loss. Our multi-class data centroid is different from Ding et al. (2022). They consider (the transpose of)

$$\mu_S^{sq} := \frac{1}{N} \sum_{i=1}^{N} y_i \psi(x_i)^T,$$

where $y_i$ is the one-hot encoding for the class. Our method involves the vector $c_{y_i}$ instead of the vector $y_i$. The relationship between the unhinged multi-class data centroid and the mean squared multi-class data centroid is given by the following equation: $\mu_S^{unh} = \mu_S^{sq} - \frac{1}{C}\left(\frac{1}{N} \sum_{i=1}^{N} \mathbf{1}\psi(x_i)^T\right)$, where $\mathbf{1}$ is a column vector with all entries equal to 1. We can think of $\mu_S^{unh}$ as a corrected version of $\mu_S^{sq}$.

## D    PROOFS

**Proof of Proposition 4.1:** Define $L^{sym}(z, y)$ by the following formula:

$$L^{sym}(z, y) := L(z, y) - \frac{1}{C} \sum_{k=1}^{C} L(z, k).$$

The loss function $L^{sym}(z, y)$ is symmetric and $L(z, y) - L^{sym}(z, y)$ is class-insensitive. This shows the existence of the decomposition. For uniqueness, suppose $L = L_1^{sym} + L_1^{ins} = L_2^{sym} + L_2^{ins}$. That is, we have two decompositions of $L$ as a sum of a symmetric and a class-insensitive loss function. Then,

$$L_1^{sym} - L_2^{sym} = L_2^{ins} - L_1^{ins}.$$

Since $L_1^{sym} - L_2^{sym}$ is symmetric and $L_2^{ins} - L_1^{ins}$ is class-insensitive, they must be both symmetric and class-insensitive. The only loss functions that are both class-insensitive and symmetric are constant. Indeed, if an arbitrary loss function $L'(z, y)$ is both symmetric and class-insensitive, then

$$constant = \sum_{y=1}^{C} L'(z, y) = CL'(z, k)$$

for any $1 \le k \le C$ and any $z \in \mathbb{R}^C$, and therefore $L'(z, y)$ is constant. We conclude that $L_1^{sym}$ is equal to $L_2^{sym}$ up to an additive constant and also $L_1^{ins}$ is equal to $L_2^{ins}$ up to an additive constant.∎

In order to prove the uniqueness result among convex functions for the multi-class unhinged loss, we start by proving uniqueness among linear functions. The invariance to permutations property is crucial and the next Lemma proves some constraints that must hold on the coefficients of a linear loss satisfying the invariance to permutations property.

**Lemma D.1** *Assume that a linear loss $L(z, y) = \sum_{k=1}^{C} a_k(y) z_k$ satisfies the invariance to permutations property. Then,*

*i) $a_y(y) = a_k(k)$ for all $k, y$.*

*ii) $a_k(y) = a_y(k)$ for all $k, y$.*

*iii) $a_k(y) = a_{k'}(y)$ if $k \ne y$ and $k' \ne y$.*

**Proof:**

    i) For given $k$ and $y$, consider a permutation $\tau$ switching $k$ and $y$. From the invariance to permutations property, $L(z, y) = L(\tau(z), \tau(y)) = L(\tau(z), k)$ for all $z$. Pick $z = e_y$. Then,

$$a_y(y) = L(e_y, y) = L(\tau(e_y), k) = L(e_k, k) = a_k(k).$$

    ii) Consider $\tau$ as above, but now pick $z = e_k$. Then,

$$a_k(y) = L(e_k, y) = L(\tau(e_y), \tau(k)) = L(e_y, k) = a_y(k).$$

    iii) Fix $y$ and let $k \ne y$ and $k' \ne y$. Consider any permutation $\tau$ with fixed point $y$ satisfying $\tau(k) = k'$ and $\tau(k') = k$. From the invariance to permutations property, $L(z, y) = L(\tau(z), \tau(y)) = L(\tau(z), y)$. Since any two linear functions equal everywhere must have the same coefficients, from the equality $L(\tau(z), y) = L(z, y)$ and noting that the coefficient of $z_k$ in $L(z, y)$ is $a_k(y)$ and the coefficient of $z_k$ in $L(\tau(z), y)$ is $a_{k'}(y)$, we conclude that $a_k(y) = a_{k'}(y)$.

**Proposition D.2** *The multi-class unhinged loss is the unique (up to a multiplicative constant) non-trivial, non-increasing, linear multi-class symmetric loss function that satisfies the property of invariance to permutations.*

**Proof:** It is easy to verify that the multi-class unhinged loss function is a symmetric, non-increasing, linear loss function that satisfies the invariance to permutations property.

We now show that it is the unique such loss function up to a multiplicative constant. Let $L(z, y) = \sum_{k=1}^{C} a_k(y) z_k$. Using the symmetry condition and rearranging terms leads to

$$\sum_{k=1}^{C} \left( \sum_{y=1}^{C} a_k(y) \right) z_k = constant,$$

for all $z$. If a linear function is constant, all its coefficients must be 0 and so

$$\sum_{y=1}^{C} a_k(y) = 0,$$

for all $k$. Using Lemma D.1 $ii)$ leads to $\sum_{y=1}^{C} a_y(k) = 0$, for all $k$. For convenience, we change the names of the indices in the previous equality and write

$$\sum_{k=1}^{C} a_k(y) = 0,$$

for all $y$. From the assumption that $L(z, y)$ is non-increasing, we must have $a_y(y) \leq 0$. If $a_y(y) = 0$, then $\sum_{k \neq y} a_k(y) = 0$. But, from Lemma D.1 $iii)$, we must then have $(C - 1)a_k(y) = 0$ for any $k$. The loss $L(z, y)$ would then be identically 0. Therefore, for the loss to be non-trivial, we must have $a_y(y) < 0$. Since we consider the loss up to a multiplicative constant, we can assume that $a_y(y) = -1$ for all $y$ (this holds for all $y$ from Lemma D.1 $i)$). Finally, from Lemma D.1 $iii)$, $a_k(y) = \frac{1}{C-1}$ if $k \neq y$. Consequently,

$$L(z, y) = -z_y + \frac{1}{C - 1} \sum_{k \neq y} z_k = \frac{C}{C - 1} \left( -z_y + \frac{1}{C} \sum_{k=1}^{C} z_k \right).$$

This concludes the proof showing that $L(z, y)$ is equal to the multi-class unhinged loss function up to a multiplicative constant.∎

**Remark D.3** *Without the property of invariance to permutations, the uniqueness result would not be true. Indeed, consider the following example with 3 classes:*

$$L(z, y) = \begin{cases} -z_1 + z_2 + z_3 & \text{if } y=1 \\ -z_2 + z_1 & \text{if } y=2 \\ -z_3 & \text{if } y=3 \end{cases}$$

*This loss function is convex (actually linear) for all $y$, non-increasing and symmetric. However, it is not equivalent to the multi-class uninged loss function.*

**Proof of Theorem 5.1:** Assume that $L(z, y)$ is a convex function of $z$. From the symmetry condition, we get

$$L(z, y) = constant - \sum_{k \neq y} L(z, k).$$

Since a sum of convex functions is convex, $-\sum_{k \neq y} L(z, k)$ is concave. It follows that $L(z, y)$ is both convex and concave. The only functions that are both convex and concave are affine functions. Therefore, under the assumptions of the theorem, it follows from Proposition D.2 that $L(y, z)$ must be equal to the multi-class unhinged loss function up to an additive and a multiplicative constant. ∎

**Proof of Proposition 5.2:** We first need to show that $\left[ \nabla_z L(z, y)|_{z=z'} \right]^T z$ satisfies the property of invariance to permutations. Let $\tau$ be a permutation and $P$ the corresponding permutation matrix. From the chain rule and the invariance to permutations property for $L(z, y)$, we get

$$\begin{aligned}
\left[ \nabla_z L(z, \tau(y))|_{z=z'} \right]^T \tau(z) &= \left[ \nabla_z L(\tau^{-1}(z), y)|_{z=z'} \right]^T \tau(z) \\
&= \left[ \nabla_z L(z, y)|_{z=\tau^{-1}(z')} \right]^T P^{-1} \tau(z) \\
&= \left[ \nabla_z L(z, y)|_{z=z'} \right]^T z,
\end{aligned}$$

where the last line is true since $\tau^{-1}(z') = z'$ when all the components of $z'$ are equal. We conclude that $\left[\nabla_z L(z,y)|_{z=z'}\right]^T z$ satisfies the property of invariance to permutations.

We now need to show that $\left[\nabla_z L(z,y)|_{z=z'}\right]^T z$ satisfies the symmetry condition. Differentiating both sides of the symmetry condition for $L(z,y)$ with respect to $z$ at $z'$ leads to

$$\sum_{k=1}^{C} \nabla_z L(z,k)|_{z=z'} = 0.$$

Taking the dot product with $z$ on both sides leads to the conclusion that $\left[\nabla_z L(z,y)|_{z=z'}\right]^T z$ is symmetric. From the uniqueness result for the multi-class unhinged loss, the proposition follows if $\left[\nabla_z L(z,y)|_{z=z'}\right]^T z$ is non-trivial, that is, if $\nabla_z L(z,y)|_{z=z'} \neq 0$. ∎

**Example D.4** *When a differentiable loss function is globally symmetric, it is also locally symmetric everywhere. However, it need not be equivalent to the multi-class unhinged loss everywhere. Indeed, even if the loss function satisfies the property of invariance to permutations, the linear approximation might not satisfy the same property everywhere. An example is the MAE in three variables:*

$$L(z,y) = 2 - 2\frac{\exp(z_y)}{\sum_{k=1}^{3} \exp(z_k)}.$$

*If we let $l(z,y) = \left[\nabla_z L(z,y)|_{z=z'}\right]^T z$ with $z' = (1,0,0)$, we get*

$$l(z,y) = \frac{2}{(e+2)^2} \begin{cases} (-2e, e, e) \cdot z & \text{if y=1} \\ (e, -e-1, 1) \cdot z & \text{if y=2} \\ (e, 1, -e-1) \cdot z & \text{if y=3} \end{cases}.$$

*This loss is symmetric as it should be. However, it does not satisfy the property of invariance to permutations and it is not equivalent to the multi-class unhinged loss function.*

**Proof of Proposition 5.3:** Denote by $R_y(z)$ the remainder for the linear approximation of $\phi(z,y)$ at 0. Then,

$$L_D(f_\phi^*) - L_D(f_L^*) = \mathbb{E}_{x,y\sim D}\left[L(f_\phi^*(x), y) - L(f_L^*(x), y)\right]$$

$$= \mathbb{E}_{x,y\sim D}\left[\phi(f_\phi^*(x), y) - \phi(0, y) - R_y(f_\phi^*(x)) - L(f_L^*(x), y)\right]$$

$$= \mathbb{E}_{x,y\sim D}\left[\phi(f_\phi^*(x), y)\right] - \mathbb{E}_{x,y\sim D}\left[\phi(0, y) + R_y(f_\phi^*(x)) + L(f_L^*(x), y)\right]$$

$$\leq \mathbb{E}_{x,y\sim D}\left[\phi(f_L^*(x), y)\right] - \mathbb{E}_{x,y\sim D}\left[\phi(0, y) + R_y(f_\phi^*(x)) + L(f_L^*(x), y)\right]$$

$$= \mathbb{E}_{x,y\sim D}\left[\phi(f_L^*(x), y) - \phi(0, y) - R_y(f_\phi^*(x)) - L(f_L^*(x), y)\right]$$

$$= \mathbb{E}_{x,y\sim D}\left[R_y(f_L^*(x)) - R_y(f_\phi^*(x))\right]$$

$$\leq 2\sup_{z,y}|R_y(z)|.$$

The proof is completed by exploiting $\beta$-smoothness to bound the remainder and by bounding the size of the outputs of the neural network:

$$|R_y(z)| \leq \tfrac{\beta}{2}||z||^2 \leq \tfrac{\beta}{2}(Rc^{(l-1)}r)^2.$$

∎

**Proof of Proposition C.1:** We have

$$
\begin{aligned}
||\mu_{S,\theta}^{unh}||_{FR}^2 &= \sum_{k=1}^{C} ||\frac{1}{N}\sum_{i=1}^{N} c_{y_i}^{(k)}\psi_\theta(x_i)||^2 \\
&= \sum_{k=1}^{C} \left( \frac{1}{N}\sum_{i=1}^{N} c_{y_i}^{(k)}\psi_\theta(x_i)^T \right) \left( \frac{1}{N}\sum_{j=1}^{N} c_{y_j}^{(k)}\psi_\theta(x_j) \right) \\
&= \frac{1}{N^2}\sum_{k=1}^{C}\sum_{i,j} c_{y_i}^{(k)} c_{y_j}^{(k)} k_\theta(x_i,x_j) \\
&= \frac{1}{N^2}\sum_{i,j} k_\theta(x_i,x_j) \left( \sum_{k=1}^{C} c_{y_i}^{(k)} c_{y_j}^{(k)} \right).
\end{aligned}
$$

The quantity $\sum_{k=1}^{C} c_{y_i}^{(k)} c_{y_j}^{(k)}$ can be easily computed. If $y_i = y_j$ then,

$$
\sum_{k=1}^{C} c_{y_i}^{(k)} c_{y_j}^{(k)} = \left( \frac{C-1}{C} \right)^2 + (C-1)\frac{1}{C^2} = \frac{C-1}{C}.
$$

If $y_i \neq y_j$ then,

$$
\sum_{k=1}^{C} c_{y_i}^{(k)} c_{y_j}^{(k)} = 2\left( \frac{C-1}{C} \right)\left( \frac{-1}{C} \right) + (C-2)\frac{1}{C^2} = \frac{-1}{C}.
$$

Let $a_{ij}$ be equal to $\frac{C-1}{C}$ if $y_i = y_j$ and $\frac{-1}{C}$ if $y_i \neq y_j$. We then get

$$
||\mu_{S,\theta}^{unh}||_{FR}^2 = \frac{1}{N^2}\sum_{i,j} a_{ij} k_\theta(x_i,x_j).
$$

∎

## E    Conditional data-dependent Dirichlet priors

In the case where the original loss function is the negative log-likelihood, our symmetrization method can be interpreted as a form of regularization induced from using Dirichlet priors on the outputs of the network. We explain precisely how below.

We consider neural networks having as outputs probability vectors on $C$ classes. That is, for a neural network with parameters $\theta$, the output of the neurak network on input $x$ is the conditional distribution $p(y|x,\theta)$. Let $\Delta_C := \{\pi = (\pi_1, \cdots, \pi_C) \mid \pi_i \geq 0 \text{ } and \text{ } \sum_{i=1}^{C} \pi_i = 1\}$ be the probability simplex in dimension $C$. A neural network $f_\theta$ is then a function

$$
f_\theta : \mathbb{R}^d \longrightarrow \Delta_C.
$$

Suppose that we have a training set of $n$ i.i.d. pairs $(x_i, y_i)$ and denote by $X$ the $d \times n$ matrix obtained from aggregating the $n$ column vectors $x_i$. Also denote by $Y$ the column vector of training labels. In a Bayesian treatment, we would be interrested in the posterior distribution $p(\theta \mid X, Y)$. From Bayes rule, we get

$$
p(\theta \mid X, Y) \propto p(Y \mid X, \theta) p(\theta \mid X).
$$

It is commonly assumed that the prior is chosen completely independently of the training data, that is $p(\theta \mid X) = p(\theta)$. However, it is also possible to maintain the dependency on $X$, leading to the notion of a data-dependent prior. This data-dependent prior depends only on the observed covariates X and not on the observed response variables $Y$.

We want to define $p(\theta \mid X)$. Each $\theta$ represents a neural network $f_\theta$. Since we have access to $X$, we can look at $f_\theta(x) \in \Delta_C$ for $x$ in the training examples to define the prior. In Bayesian

statistics, the Dirichlet distribution of order $C$ is often used as a distribution over $\Delta_C$ since it is the conjugate prior to the categorical distribution. It is defined by the density function

$$g(\pi\,;\alpha_1,\cdots,\alpha_C) = constant \times \prod_{i=1}^{C} \pi_i^{\alpha_i-1},$$

where the parameters $\alpha_i$ satisfy $\alpha_i > 0$ for all $i$. If $\pi \in \Delta_C$ is distributed according to a Dirichlet distribution with parameters $\alpha = (\alpha_1,\cdots,\alpha_C)$, we will write $\pi \sim Dir(\alpha)$. A very natural first step to define our data-dependent prior $p(\theta \,|\, X)$ is to let $f_\theta(x_i) \sim Dir(\alpha(x_i))$ for each $x_i$ in the training set and where $\alpha(x)$ is a function of $x$. We then need to define a joint distribution over the vector $(f_\theta(x_1),\cdots,f_\theta(x_n))$. We will simply choose to have the $f_\theta(x_i)$'s mutually independent. We then have a joint distribution over the outputs of $f_\theta$ on the training data. This does not lead immediately to a distribution on $\theta$ however since many different $\theta$'s can have the same vector of outputs $(f_\theta(x_1),\cdots,f_\theta(x_n))$. We define an equivalence class on $\theta$ as follows:

$$[\theta]_X = [\theta']_X \text{ if and only if } f_\theta(x_i) = f_{\theta'}(x_i) \text{ for all } 1 \le i \le n.$$

So far, we have defined a distribution $p([\theta]_X \,|\, X)$. It is given by a product of Dirichlet distributions (to be technically correct, we have to take the restriction of this product of Dirichlet to the subset of $\Delta_C^n$ that can be realized with the hypothesis class $\{f_\theta\}$). We can then write

$$p(\theta \,|\, X) = \int_{[\theta']_X} p(\theta \,|\, [\theta']_X, X)p([\theta']_X \,|\, X)d[\theta']_X = p(\theta \,|\, [\theta]_X, X)p([\theta]_X \,|\, X).$$

We are therefore left with defining a distribution for $\theta$ inside of its equivalence class i.e. $p(\theta \,|\, [\theta]_X, X)$. A possible example would be to choose a uniform distribution. This would lead to $p(\theta \,|\, [\theta]_X, X) = \frac{1}{m([\theta]_X)}$, where $m([\theta]_X)$ is the measure of the set $\{\theta' \, s.t. \, [\theta']_X = [\theta]_X\}$. Up to an additive constant the negative log posterior is then given by

$$\sum_{i=1}^{n} -\log(p(y_i \,|\, x_i, \theta)) + \sum_{i=1}^{n}\sum_{k=1}^{C} -(\alpha_k(x_i) - 1)\log(p(k \,|\, x_i, \theta)) - \log(p(\theta \,|\, [\theta]_X, X)).$$

If we denote by $l(h(x), y)$ the negative log likelihood loss ($l(h(x), y) = -\log(p(y \,|\, h(x)))$, where $h(x)$ is the score vector) and if we drop the extra regularization term $-\log(p(\theta \,|\, [\theta]_X, X))$, then the quantity above is the sum over all examples of

$$l_{Dir}(h(x), y) := l(h(x), y) + \sum_{k=1}^{C}(\alpha_k(x) - 1)l(h(x), k).$$

**Lemma E.1** *Assume that $\alpha_k(x) = \alpha$ is constant. Then, the loss $l_{Dir}$ is symmetric if and only if $\alpha = \frac{C-1}{C}$ or $l$ is already symmetric.*

**Proof:** If $\alpha_k(x) = \alpha$, we have

$$\sum_{y=1}^{C} l_{Dir}(h(x), y) = \left[1 + C(\alpha - 1)\right]\sum_{y=1}^{C} l(h(x), y).$$

Therefore, if $l$ is not already symmetric, we must have $1 + C(\alpha - 1) = 0$ for $l_{Dir}$ to be symmetric. That is, we must have $\alpha = \frac{C-1}{C}$.∎

The special case of $l_{Dir}$ with $\alpha_k(x) = \frac{C-1}{C}$ leads to the same loss as $l^{sym}$, that is, the symmetrization of $l$. An illustration for the binary case (Beta distributions) is given in Figure 1. The added prior encourages more confident outputs.

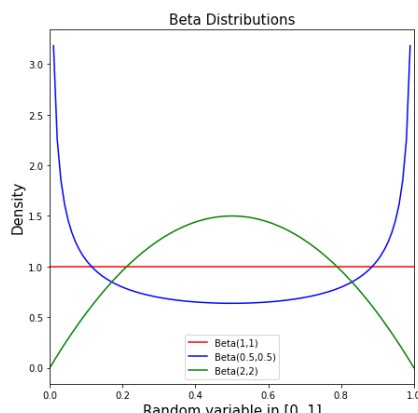

Figure 1: The added regularization (induced from a $Beta(\frac{1}{2}, \frac{1}{2})$ prior in the binary case) favours probability vectors with less entropy.

## F    REGRESSION

The present paper focused on extending decomposition, symmetrization and the binary unhinged loss function to the multi-class case. It is also possible to extend these ideas to regression. Suppose that we are now trying to predict a continuous variable $y \in \mathbb{R}$ and that the data collection process is noisy. The training data comes from a different distribution (corrupted distribution, cheaper to obtain samples from) than the test data (clean distribution). A formalization is given below:

**Definition F.1** *Consider the problem of regression. Define the corrupted distribution $\overline{D}$ to have the same marginal distribution as $D$ (that is $\overline{D}_x = D_x$) but with conditional distribution $\overline{D}_{y|x}$ given by*

$$pq_x(y) + (1-p)D_{y|x},$$

*where $0 \leq p < 1$ and $q_x(y)$ is the corruption distribution. Two examples are $\mathcal{N}(\mu_x, \sigma_x^2)$ (a Gaussian distribution with mean $\mu_x$ and variance $\sigma_x^2$) and a uniform distribution (when the domain of $y$ is a bounded interval in $\mathbb{R}$).*

**Lemma F.2** *Let $q_x(y)$ be the density for the corruption distribution. Define $\gamma_h(x) = \int q_x(y)l(h(x), y)dy$ and $\Gamma_h = \mathbb{E}_D \gamma_h(x)$. For any $h \in \mathcal{H}$,*

$$L_{\overline{D}}(h) = p\Gamma_h + (1-p)L_D(h).$$

**Proof:** We have

$$L_{\overline{D}}(h) = p\left[\mathbb{E}_{x \sim D_x} \int q_x(y)l(h(x), y)dy\right] + (1-p)L_D(h) \qquad (5)$$

$$= p \mathbb{E}_{x \sim D_x} \gamma_h(x) + (1-p)L_D(h). \qquad (6)$$

**Definition F.3** *We say that a regression loss function satisfies the continuous symmetry condition with respect to density $q_x(y)$ if*

$$\int q_x(y)l(h(x), y)dy = constant.$$

**Corollary F.4** *If $l$ satisfies the continuous symmetry condition with respect to $q_x(y)$ then minimizing $L_{\overline{D}}(h)$ is equivalent to minimizing $L_D(h)$.*

There is also a unique decomposition of any regression loss function as a sum of a $q_x(y)$-symmetric loss function and a label-insensitive term. The proof is almost identical to the

proof for classification and is omitted. Consider the simpler case where $q_x(y) = q(y)$ is independent of $x$. Define $L^{sym}(z, y)$ by the following formula:

$$L^{sym}(z, y) = L(z, y) - \int q(y)L(z, y)dy.$$

Then $L^{sym}(z, y)$ satisfies the continuous symmetry condition with respect to $q(y)$ and is the symmetric loss function in the unique decomposition of $L$. The closest generalization of the classification case is to take a uniform distribution on a bounded domain $[-I, I]$. We investigate this case in conjunction with the squared error loss in the example below.

**Example F.5** *Consider the squared error loss $L(z, y) = (z - y)^2$. A direct computation leads to a loss equivalent (up to an additive and a multiplicative constant) to $-zy$ for $L^{sym}(z, y)$. We could refer to this loss as the regression unhinged loss function. It is unbounded and would suffer from numerical overflows without proper regularization. In the case of linear classifiers, we can give an explicit solution. The regularized objective (with l2-regularization) is given by*

$$\frac{1}{N} \sum_{i=1}^{N} -y_i w\psi(x_i) + \frac{\lambda}{2}||w||_2^2.$$

*Here, $w$ is a row vector and $\psi(x_i)$ is a column vector after applying a feature map $\psi$ to $x_i$. Finding when the gradient is $0$ leads to:*

$$w = \frac{1}{\lambda N} \sum_{i=1}^{N} y_i\psi(x_i)^T.$$

*This means that $w$ depends only on the data centroid (for regression)*

$$\mu_S = \frac{1}{N} \sum_{i=1}^{N} y_i\psi(x_i)^T$$

*and a scaling factor $\frac{1}{\lambda}$. Interestingly, the linear approximation at $0$ of the squared error loss is also the regression unhinged ($-2yz$).*

If $L(z, y)$ is symmetric, the linear approximation at any points $z'$ is also symmetric as can be shown by differentiating both sides of the symmetry condition and multiplying by $z$. We only need to differentiate under the integral for this result to be true, which can be done if we assume that, for example, $q(y)L(z, y)$ and its derivative with respect to $z$ are continuous in $z$ and $y$. It is also true in the regression case that a symmetric and convex loss function must be affine.

**Proposition F.6** *Assume that $L(z, y)$ is a twice differentiable function of $z$ for any $y$ and that the second derivative is continuous in $z$ and $y$. If $L(z, y)$ is symmetric and convex for any $y$ then it is an affine function of $z$ for any $y$.*

**Proof:** Differentiating under the integral twice with respect to $z$ in the symmetry condition leads to

$$\int q(y)L''(z, y)dy = 0,$$

where $L''(z, y)$ denotes the second derivative with respect to $z$. Since, $L(z, y)$ is convex for any $y$, we must have $L''(z, y) \geq 0$ for any $z$ and $y$. Since the integral above is $0$, it follows that $L''(z, y)$ is identically $0$. We conclude that $L(z, y)$ is an affine function of $z$ for any $y$. ∎

A linear regression loss (from the point of view of $z$) is of the form $f(y)z$ for some function $f$. Such a loss function is symmetric if and only if

$$\mathbb{E}_{y \sim q(y)} f(y) = 0.$$

We have shown in this section that the theory of symmetric loss functions can be extended to regression to a large extent with very similar results to classification.

Table 5: Hyperparameters and training configuration for SGCE.

|  | CIFAR10 (Tables 1,2) | CIFAR100 (Tables 1,2) | CIFAR100 (Table 4) | WebVision (Table 3) |
|---|---|---|---|---|
| q | 0.80 | 0.65 | 0.35 | 0.25 |
| train batchsize | 128 | 128 | 128 | 32 |
| total epoch | 120 | 200 | 200 | 250 |
| optimizer | SGD | SGD | SGD | SGD+Nesterov |
| learning rate | 0.01 | 0.1 | 0.1 | 0.1 |
| momentum | 0.9 | 0.9 | 0.9 | 0.9 |
| weight decay | 0.005 | 0.0005 | 0.0005 | 0.00005 |
| gradient bound | 5.0 | 5.0 | 5.0 | 5.0 |
| scheduler | cosine | cosine | steplr | steplr |
| T_max | 120 | 200 | N/A | N/A |
| eta_min | 0.0 | 0.0 | N/A | N/A |
| step_size | N/A | N/A | 190 | 240 |
| gamma | N/A | N/A | 0.1 | 0.1 |

Table 6: Hyperparameters and training configuration for $\alpha$-MAE.

|  | CIFAR10 (Tables 1,2) | CIFAR100 (Tables 1,2) | CIFAR100 (Table 4) |
|---|---|---|---|
| $\alpha$ | 2.0 | 2.0 | 0.25 |
| train batchsize | 128 | 128 | 128 |
| total epoch | 120 | 200 | 200 |
| optimizer | SGD | SGD | SGD |
| learning rate | 0.01 | 0.1 | 0.1 |
| momentum | 0.9 | 0.9 | 0.9 |
| weight decay | 0.005 | 0.0005 | 0.0005 |
| gradient bound | 5.0 | 5.0 | 5.0 |
| scheduler | cosine | cosine | steplr |
| T_max | 120 | 200 | N/A |
| eta_min | 0.0 | 0.0 | N/A |
| step_size | N/A | N/A | 190 |
| gamma | N/A | N/A | 0.1 |

## G  TRAINING CONFIGURATION AND HYPERPARAMETERS

The hyperparameters used in order to obtain our results are given in Table 5 (SGCE), Table 6 ($\alpha$-MAE) and Table 7 (multi-class unhinged). The scheduler "steplr" refers to the scheduler torch.optim.lr_scheduler.StepLR with parameters "step_size" and "gamma". The scheduler "cosine" refers to the scheduler torch.optim.lr_scheduler.CosineAnnealingLR with parameters "T_max" and "eta_min". The weight decay parameters for the different loss functions are given in Table 8.

Table 7: Hyperparameters and training configuration for the multi-class unhinged loss function.

| | CIFAR10 (Tables 1,2) | CIFAR100 (Tables 1,2) | CIFAR100 (Table 4) |
|---|---|---|---|
| train batchsize | 128 | 128 | 128 |
| total epoch | 120 | 200 | 200 |
| optimizer | SGD | SGD | SGD |
| learning rate | 0.01 | 0.1 | 0.1 |
| momentum | 0.9 | 0.9 | 0.9 |
| weight decay | 0.01 | 0.001 | 0.0005 |
| gradient bound | 5.0 | 5.0 | 5.0 |
| scheduler | cosine | cosine | steplr |
| T_max | 120 | 200 | N/A |
| eta_min | 0.0 | 0.0 | N/A |
| step_size | N/A | N/A | 190 |
| gamma | N/A | N/A | 0.1 |

Table 8: Weight Decay Parameters for Different Loss Functions and Datasets

| Loss Function | CIFAR10 | CIFAR100 (cosine) | CIFAR100 (steplr) |
|---|---|---|---|
| SGCE | 0.005 | 0.0005 | 0.0005 |
| $\alpha$-MAE | 0.005 | 0.0005 | 0.0005 |
| Unhinged | 0.01 | 0.001 | 0.0005 |
| CE | 0.005 | 0.001 | 0.0005 |
| MAE | 0.0001 | 5e-5 | None |
| GCE | 0.005 | 0.001 | 0.0001 |
| SCE | 0.01 | 0.0005 | None |
| NCE+RCE | 0.0001 | 1e-5 | None |
| NCE+AGCE | 0.0001 | 1e-5 | None |
| ANL-CE | 0.0 | 0.0 | 1e-5 |

