# OpenReview forum: "Symmetrization of Loss Functions for Robust Training of Neural Networks in the Presence of Noisy Labels"
_ICLR.cc/2025/Conference — Submitted to ICLR 2025_

### Official Review · Reviewer_Vrd9 · 2024-10-21

**Soundness:** 2
**Presentation:** 3
**Contribution:** 2
**Rating:** 5
**Confidence:** 4

**Summary:**

In this paper,  a symmetrization method for designing robust loss functions to handle label noise is proposed as mainly illustrated in Section 4. this loss function.  This may be regarded as a good extension of the binary unhinged loss in the multi-class case. The symmetrization of the generalized crossentropy loss (SGCE) and the  alpha-MAE  are introduced with some experimental validation.

**Strengths:**

1. The paper introduced  a symmetrization method for designing robust loss functions to handle label noise in the multi-class setting.  It is a good extension of the unhinged loss in binary case to multi-class cases.

2. Various properties of the multi-class unhinged loss are discussed.

**Weaknesses:**

1. Many of the properties of the multi-class unhinged loss function seem to be straightforward and may not offer substantial novelty.  I have not checked very step of the proofs and was not able to judge how significant the proof techniques are.

2. While the introduction of the symmetrization method for designing robust loss functions in the multi-class setting is intriguing, the results feel somewhat incremental and lack sufficient depth. For instance, in the binary case, there are established results in binary classification that show how the design of robust loss functions to handle label noise relates to statistical consistency. I would encourage the authors to exploring this connection explicitly which will increase the impact and significance of the paper.

3. The presentation could be improved. For example, immediately after Definition 3.1, the authors mention that the form of symmetry between classes differs from the notion of symmetry related to noise robustness. However, in Section 4, the paper states, "to decompose a loss function into a sum of a symmetric loss function and a class-insensitive term," yet the definition of a symmetric loss function has not been provided.  I would suggest the authors to clearly define the symmetric loss function before Section 4.

The writing seems to assume that readers already know very well the literature on robust (symmetric) loss functions for handling label noise. For instance, there are various key terms such as robust loss functions based on the symmetry condition, symmetric loss function, symmetrization of the cross-entropy loss, unhinged loss.  It would be beneficial to the readers to properly define and introduce them in a more logical order before mentioning them.

**Questions:**

See the weakness part

---

> ### Author Response · Authors · 2024-11-28
> **Thank you for your time and comments**
>
> 1) We appreciate your feedback regarding the novelty of our approach. In the following paragraph, we clarify the position of our work in relation to previous literature to better highlight the novelty of our contributions.
>
> In [1], a general method for producing symmetric loss functions from non-symmetric loss functions, referred to as normalization, was proposed. However, when applying normalization to the cross-entropy loss, the resulting loss (NCE) could not be trained successfully on CIFAR100 due to severe underfitting. They then proposed to linearly combine two symmetric loss functions (essentially $\alpha$ NCE+$\beta$ MAE), and the resulting combination could be trained successfully after proper tuning of the two hyperparameters. The subsequent work of [2] replaced the MAE with a different loss (normalized negative loss) to boost the results. In this work, we propose a different general operation referred to as symmetrization for producing new symmetric loss functions. The idea behind the symmetrization operation is that there is a "hidden" symmetric loss function inside of any loss function (a symmetric component of the loss function), as shown by the result on unique decomposition. When applied to the cross-entropy loss, the resulting loss is an interesting mathematical object (the simplest symmetric loss; the unhinged), while the normalization operation produces a loss function with no special mathematical interest in our understanding. We believe our approach to be mathematically principled and empirically effective, as shown with our results. Additionally, we introduce the property of invariance to permutations (Definition 3.2), which plays a key role in the theory of multi-class symmetric loss functions (Theorem 5.1 and Proposition 5.2).
>
>  2) Statistical consistency is also established in the multi-class case not only in the binary case [3]. Our loss functions therefore does satisfy this property.
>
>  3) The definition of symmetry is provided in Definition A.1. To improve clarity, we added the definition also directly in the introduction.
>
>
> [1]: Normalized loss functions for deep learning with noisy label, Xingjun Ma, Hanxun Huang, Yisen Wang, Simone Romano, Sarah Erfani, and James Bailey, ICML, 2020.
>
> [2]: Active negative loss functions for learning with noisy labels. Xichen Ye, Xiaoqiang Li, songmin dai, Tong Liu, Yan Sun, and Weiqin Tong, Neurip, 2023.
>
> [3]: Aritra Ghosh, Himanshu Kumar, and P. S. Sastry. Robust loss functions under label noise for deep neural networks. In Proceedings of the Thirty-First AAAI Conference on Artificial Intelligence, AAAI’17, page 1919–1925. AAAI Press, 2017.

---

### Official Review · Reviewer_PRzr · 2024-10-30

**Soundness:** 2
**Presentation:** 2
**Contribution:** 2
**Rating:** 5
**Confidence:** 5

**Summary:**

This paper focuses on the robust loss functions in noisy label learning. Since the symmetric loss is proved to be robust to noisy labels, the authors propose a method to symmetrize the loss function. The corresponding theoretical analysis is provided, including a proof that the multi-class unhinged loss is the only convex multi-class symmetric loss. The authors use three loss functions, and evaluate on CIFAR-10, CIFAR-100 and WebVision.

**Strengths:**

1. The authors use a  clever way to achieve symmetric condition.
2. This paper provides a reasonable theoretical analysis.
3. The authors make comparisons on different noises, including symmetric noise, asymmetric noise, human-annotated noise, and real world noise.
4. Three new robust loss functions are proposed.

**Weaknesses:**

1. The fonts used don't seem to conform to the ICLR format. Futhermore, I think the symbol definitions are confusing. The lowercase characters in the paper can represent both vectors and constants, as well as elements of vectors. A clearer distinction would have been better.
2. Although the authors' method implements the symmetric condition, I noticed that in the actual experiment the authors used a lot of tricks, including Euclidean normalization and batch normalization for $z$, and weight decay adjustment. However, baselines[1, 2] do not need to use these tricks.
3. The authors did not perform experiments according to the uniform experimental setting of baselines[1, 2]. Instead, they search for optimal parameters and optimal weight decay for their own method, but only for weight decay without optimal hyperparameters for baselines. A fairer way to experiment is to refer to [3], or to use the uniform experimental setting of baselines[1, 2].
4. No code is provided for this work. It would be better to provide code about Euclidean normalized and batch normalized multi-class unhinged loss, etc.

**Questions:**

1. The author mainly analyzes multi-class unhinged loss in theory, but uses Euclidean normalized and batch normalized multi-class unhinged loss in practice. After using normalization, does the proposed theory still hold？
2. Is there any corresponding theoretical analysis about the use of  Euclidean normalization and batch normalization？
3. Since the unhinged loss is already symmetric and convex, why is there a trade-off between it and MAE? Why does such an operation improve performance? Is there any theoretical analysis？
4. Since SGCE and $\alpha$-MAE are both a trade-off between unhinged and MAE, is there a theoretical explanation for why $\alpha$-MAE performs better in most cases?
5. Can you provide the results in the same experimental setting as baselines [1, 2] ?

[1] Normalized Loss Functions for Deep Learning with Noisy Labels, ICML 2020.

[2] Asymmetric Loss Functions for Learning with Noisy Labels, ICML 2021.

[3] Generalized Jensen-Shannon Divergence Loss for Learning with Noisy Labels, NeurIPS 2021.

---

> ### Author Response · Authors · 2024-11-28
> **Thank you for your time reading our paper and your comments**
>
> - Discussion about the experimental setting: The experimental protocol was chosen based on the more recent paper [1]. For training their loss functions, [1] are tuning 3 hyperparameters.  One hyperparameter $\delta$ for regularization (they actually use l1-regularization since they claim it performed better than l2-regularization) and 2 hyperparameters for the loss. In comparison, we tune also 1 parameter for regularization (but stick to l2-regularization) and only 1 parameter for the loss. This places us at a disadvantage in terms of tuning compared to [1], yet our results still compare favorably. There is also another very important factor involved for the loss functions from the active passive framework [2], [1]. These loss functions combine two loss functions using two hyperparameters $\alpha$ and $\beta$. Some combinations of $\alpha$ and $\beta$ are leading to equivalent loss functions up to a multiplicative constant. This allows to implicitly tune the learning rate. Indeed, the multiplicative factor simply multiplies the gradient by a constant which is equivalent to changing the learning rate in gradient descent optimization. As an example, consider the three tuples $(\alpha,\beta): (0.1,0.1), (1.0,1.0), (10.0,10.0)$ in the grid search for the hyperparameters in [1] and [2]. This allows to implicitly consider three different learning rates. In contrast, our method does not utilize an additional hyperparameter for implicit learning rate retuning. Overall, we tune the same number of hyperparameters as NCE+RCE [2], one fewer than ANL-CE [1], and two fewer than NCE+AGCE [3].
>
> - When using batch normalization or euclidean normalization, one could see those operations as part of the architecture. Then, the vector $z$ is the ouput after the normalization operation. Results about the loss functions L(z,y) themselves are then not impacted.
>
> - The trade-off between the unhinged loss and MAE allows to control the amount of saturation in the loss ($\beta$-smoothness). This enables different trade-offs between focusing on worst-case margins and the average margin.
>
> - One hypothesis on why $\alpha$-MAE may outperform SGCE is that $\alpha$-MAE ($\alpha\in [0,\infty)$) can realize a larger range of values of $\beta$-smoothness compared to SGCE.
>
>
>
>
>
>
> [1]: Active negative loss functions for learning with noisy labels. Xichen Ye, Xiaoqiang Li, songmin dai, Tong Liu, Yan Sun, and Weiqin Tong, NeurIPS, 2023.
>
> [2]: Normalized loss functions for deep learning with noisy label, Xingjun Ma, Hanxun Huang, Yisen Wang, Simone Romano, Sarah Erfani, and James Bailey, ICML, 2020.
>
> [3] Asymmetric Loss Functions for Learning with Noisy Labels, ICML 2021.

---

> > ### Comment · Reviewer_PRzr · 2024-12-01
> >
> > Dear authors
> >
> > Thank you for your reply. However, the authors did not provide results under the same experimental settings as the baselines [1, 2]. Therefore, I will maintain my rating.
> >
> > [1] Normalized Loss Functions for Deep Learning with Noisy Labels, ICML 2020.
> >
> > [2] Asymmetric Loss Functions for Learning with Noisy Labels, ICML 2021.
> >
> > Best regards
> >
> > The review

---

### Official Review · Reviewer_iqHq · 2024-11-02

**Soundness:** 2
**Presentation:** 2
**Contribution:** 2
**Rating:** 3
**Confidence:** 4

**Summary:**

This paper proposes a symmetrization method for multi-class loss functions, which can decompose a loss function into a sum of a symmetric loss function and a class-insensitive term. Based on this method, new robust loss functions (SGCE, $\alpha$-MAE) are proposed. The authors also provide theoretical analyses of multi-class unhinged loss. To verify the effectiveness of the proposed loss functions, the authors conduct experiments on several benchmark datasets (CIFAR-10, CIFAR-100, and WebVision).

**Strengths:**

1. The research topic, training neural networks in the presence of noisy labels, is interesting.
2. The author conducted sufficient experiments to verify the effectiveness of the proposed method.

**Weaknesses:**

1. Some definitions of crucial concepts are important for this paper, such as symmetry condition (proposed in [1]). The authors did not define them clearly in the paper.
2. A claim is wrong in Lines (184-185). The standard cross-entropy loss is not permutation-invariant. For example, the predition of network is $z=[0.2,0.5,0.3]$, the one-hot target is $y=[0,1,0]$, the corresponding cross-entropy loss is $L(z,y)=-\log(z_2)=-\log(0.5)$. Now, applying the permutation $\tau(1)=3,\tau(2)=1,\tau(3)=2$. The permuted predition is $\tau(z)=[z_3,z_1,z_2]=[0.3,0.2,0.5]$, the permuted target is $\tau(y)=[1,0,0]$, the cross-entropy loss after permutation is $L(z,y)=-\log(\tau(z)_1)=-\log(0.3)$. Obviously, $-\log(0.3)\neq-\log(0.5)$. Similarly, the linear loss: $L(z,y)=-z_y$ is not permutation-invariant.
3. The proposed method in Proposition 4.1 changes the original loss: $L^{sym}(z,y):=L(z,y)-\frac{1}{C}\sum_{k=1}^CL(z,k)$. Can the author prove whether the minimizer of $L^{sym}(z,y)$ is the same as that of $L(z,y)$?
4. The theoretical analyses in Section 5 are about the multi-class unhinged loss. The readers could be more interested in the properties of the loss $L^{sym}(z,y)$.
5. The writing and organization of this paper are poor. For example, the introduction lacks sufficient detail, and the related work is overly detailed and lengthy.
6. The location of the experiment results (Tables 1-4) is inappropriate. These tables should be in the experiment section or in the front of the experiment section.

**References**

[1] Ghosh, Aritra, Himanshu Kumar, and P. Shanti Sastry. "Robust loss functions under label noise for deep neural networks." *Proceedings of the AAAI conference on artificial intelligence*. Vol. 31. No. 1. 2017.

**Questions:**

Please see the Weaknesses.

---

> ### Author Response · Authors · 2024-11-19
> **Thank you for your time reading our paper**
>
> - The definition of the symmetry condition is given in Definition A.1 (Appendix). We point to it in lines 049-050 of the introduction. We have also included the definition directly in the introduction in the updated version of the PDF. If you think important definitions are missing, we thank you for indicating which ones and we will add them.
> - The claim in lines 184-185 is not wrong. The problem in your counter example is that the permutation $\tau$ is not being applied as in Definition 3.2. We should have $\tau (z) = (z_{τ ^{−1}(1)}, · · · , z_{τ^{ −1}(C)})=(z_2,z_3,z_1)=(0.5, 0.3, 0.2)$. On the other hand, $\tau(y)=\tau(2)=1$. Then, $L(\tau(z),\tau(y))=-log(\tau(z)_1)=-log(z_2)=-log(0.5)=L(z,2)$.
> - In general, the minimizers of $L^{sym}(z,y)$ and $L(z,y)$ may differ. For instance, in the binary case, training with the logistic loss and the unhinged loss can yield different solutions. The intuition is that the unhinged loss penalizes the margins of data points linearly, while the logistic loss emphasizes penalizing data points with the worst case margins more heavily compared to the unhinged loss.
> Additional hypothesis are needed in order to obtain that the minimizers would be the same. One example of a scenario where the result would be true is when the point-wise minimal value of the loss can be reach on the whole training set $S$. More formally, assume
>
> (1)$\displaystyle$ for all $y$, $argmin_{z} L(z,y) = argmax_{z}\sum_{k\neq y}L(z,k)=\bar{z}$
>
> (2) $\displaystyle$ $min_{h\in H}L_S(h)=L(\bar{z},y)$.
>
> If (1) and (2) are satisfied then the minimizers of the empirical loss for $L^{sym}(z,y)$ and $L(z,y)$ are the same for training set $S$ on hypothesis class $H$.
>
> - Thank you for pointing out the inappropriate location for Tables 1-4, we will make the correction.

---

> > ### Comment · Reviewer_iqHq · 2024-12-03
> >
> > Dear authors,
> >
> > Thank you for your response and revision. However, the overall writing quality remains insufficient. The structure and organization of the paper need to be adjusted to enhance clarity and coherence. Specifically, the introduction can contain several paragraphs to state the background, motivation, the problem this paper solves, and the advantages/benefits of the method. For example, the author can state the motivation for designing new symmetric loss functions and the advantages of the new symmetric loss functions compared to the existing ones. These writing methods are common in previously accepted ICLR.
> >
> > The related work is lengthy and chaotic. An approach to writing is dividing the related work into several categories and explaining their differences. An explanation for the connection and difference between the existing methods and the proposed method can be included. No need to provide too much detail.
> >
> > Best regards,
> >
> > The review

---

> > > ### Author Response · Authors · 2024-12-03
> > >
> > > You rejected the paper for fundamental points that you "misunderstood" and we have re-explain and justify these points in our rebuttal. Now you bring out editorial aspects that were not raised in your review the first time and that can be easily addressed in a final version; a work of many years cannot be rejected this way.

---

### Official Review · Reviewer_mXQK · 2024-11-03

**Soundness:** 2
**Presentation:** 1
**Contribution:** 2
**Rating:** 5
**Confidence:** 4

**Summary:**

The manuscript proposes an novel symmetrization method for robust learning in the presence of label, derived from the decomposition of any multi-class loss function into a sum of a symmetric loss function and a class-insensitive term. The authors introduce the SGCE and $\alpha$-MAE functions, which provide a smooth interpolation between the multi-class unhinged loss and the Mean Absolute Error (MAE). Overall, the paper presents a straightforward and effective method for symmetrizing any loss functions, contributing both interesting theoretical insights and experimental validation of the proposed method. However, I feel that significant writing issues detract from the clarity and accessibility of the content. Reorganizing the presentation of results could enhance the paper’s readability and overall acceptability.

**Strengths:**

- This paper presents a simple and efficient method for symmetrizing loss functions to handle label noise.
- The experimental results are comprehensive and support for the proposed method.

**Weaknesses:**

- The writing quality is insufficient and may not meet the standards required for acceptance.
- The authors should give the definition of $\tau(y)$ for $y\in\{1,...,C\}$ in Definition 3.2, as $y$ does not have $C$ elements.
- A definition of one-hot vector $e_y$ is missing in Lines 224-225.
- The authors’ statement in Lines 238-241, claiming that “the unhinged loss function is robust by maintaining larger gradients for correctly classified examples, while MAE is robust by reducing the gradient of incorrectly classified examples,” requires further clarification. Both methods, in fact, focus on increasing  $z_y$  while reducing $z_k$ for $k\neq y$., so an explanation for this distinction is warranted.
- The symmetrization of the cosine similarity loss could be interpreted as an equivalent form of MAE when the softmax layer is replaced by an $\ell_2$-normalization layer.
- “The unhinged” should likely be revised to “The unhinged loss” in Lines 248-249 and Lines 250-251.
- Why using $s(z)-y$ in Lines 277-278 instead of $s(z)-e_y$?

**Questions:**

Some questions relating to the weaknesses:
- How does the symmetrization method handle cases where class imbalances may impact the performance of the symmetric loss function?
- Could the authors elaborate on the decision to use SGCE and $\alpha$-MAE specifically for interpolation between multi-class unhinged loss and MAE? Are there alternative interpolative methods that could be considered?
- How does this approach compare in terms of computational complexity to existing symmetrization techniques for multi-class loss functions?
- In practical applications, how sensitive is the proposed symmetrization method to the choice of hyperparameters, and are there any guidelines for optimal tuning?
- Could the theoretical results be more clearly linked to the observed experimental outcomes to strengthen the argument for the method's effectiveness?

---

> ### Author Response · Authors · 2024-11-25
> **Thank you for your time and comments on our paper**
>
> - Clarifications for notations: $y$ is a number in  \{$1,\cdots, C $\} as mentioned in line 168. $\tau$ is a permutation on the set \{$1,\cdots, C$\} and it acts on vectors $z$ as defined in Definition 3.2. In line 277-278, it should indeed be  $s(z)-e_y$ instead of $s(z)-y$.
> - Clarifications for the statement in Lines 238-241: For MAE, the gradient of correctly classified examples will decay to $0$ as these examples become better and better classified. Since the unhinged loss is linear, it maintains a higher gradient for these examples in comparison to MAE. For incorrectly classified examples, the gradient for MAE will be smaller in comparison to the gradient for the unhinged loss because MAE saturates.
>  - Thank you for your question regarding class imbalances. While we did not specifically address this issue in our paper, we can suggest a few potential directions for future investigation.
>  1)  Symmetrized Weighted Loss Functions: A promising approach could be the symmetrization of weighted loss functions of the form $w_yL(z,y)$, where $w_y$​ adjusts the weight for class $y$. This would aim to balance the contributions of different classes in the optimization process.
>  2) Combining Over-Sampling with Symmetric Loss Functions: Another idea is to combine minority over-sampling techniques, such as  [1], with symmetric loss functions. In scenarios where class imbalance in the training set is caused by a highly asymmetric noise rate—resulting in many examples being corrupted toward majority classes while leaving others underrepresented—the minority classes may naturally contain a higher proportion of clean examples. In such cases, over-sampling the minority classes could not only address class imbalance but also help counteract label noise. This approach may synergize with symmetric loss functions.
> - The reason for considering SGCE is that it can be obtain directly from GCE an already well established loss function. SGCE, in some sense, solves a weakness in GCE since GCE is not symmetric. $\alpha$-MAE is considered since it enables directly to control the beta-smoothness of the loss in a simple way. Temperature scaling in MAE could be an alternative. Also, MAE could be replace by any non-linear symmetric loss function in section $6$.
> - In terms of complexity, the unhinged loss is clearly the simplest. In practice, the overall computation time will mostly depend on the amount of tuning of the hyperparameters is the loss. In comparison, to the previous work of [2] our loss functions involve tuning only 2 hyperparameters instead of 3. The grid search in our case is 5x5 instead of 5x5x5.
> - In practice, the performance is indeed sensitive to hyperparameters tuning. For example, the results for the unhinged are quite different from the optimal results for SGCE and $\alpha$-MAE on CIFAR100 showing that a large range of performance can be obtained with different hyperparameters (since the unhinged is a special case of both SGCE and $\alpha$-MAE).
>
>
> [1] Chawla, Bowyer, Hall, Kegelmeyer, SMOTE: Synthetic Minority Over-sampling Technique, Journal of Artificial Intelligence Research 16 (2002)
>
> [2] Ye, Li, Dai, Liu, Sun, and Tong. Active negative loss functions for learning with noisy labels. Neurips 2023.

---

### Official Review · Reviewer_pQfd · 2024-11-03

**Soundness:** 3
**Presentation:** 2
**Contribution:** 3
**Rating:** 6
**Confidence:** 4

**Summary:**

he paper introduces a new approach for designing robust loss functions for neural network training in the presence of label noise. It proposes a symmetrization method to create symmetric loss functions, yielding novel loss functions such as SGCE and α-MAE. These loss functions demonstrate competitive robustness across various noisy label scenarios, performing well on standard benchmarks.

**Strengths:**

1- Provides a comprehensive theoretical foundation for the symmetrization method, demonstrating its potential robustness in noisy label environments.

2- Introduces two novel loss functions, SGCE and α-MAE, with empirical validation across multiple datasets and noise types, showing competitive or superior performance.

3- Addresses a key problem in deep learning, where noisy labels can lead to model memorization, with a focus on both theoretical and practical robustness.

4- The introduction of SGCE and α-MAE loss functions showcases a balance between robustness (inspired by the MAE) and performance (closer to Cross-Entropy). This aligns well with existing literature that often emphasizes a trade-off between robustness (e.g., using MAE or its variants) and fitting capacity (e.g., Cross-Entropy), as seen in works by Zhang and Sabuncu (2018) and Wang et al. (2019). However, SGCE and α-MAE offer smooth interpolations between these behaviors with fewer tuning parameters. By allowing practitioners to adjust robustness dynamically, these losses contribute an innovative middle ground not fully achieved by either purely robust or purely performance-driven loss functions in prior research.

**Weaknesses:**

1- Ambiguity in Theorem 5.1's Uniqueness Proof for Multi-Class Symmetric Loss Functions
The paper claims the uniqueness of the multi-class unhinged loss as the sole convex, non-trivial, non-increasing multi-class symmetric loss function. However, the proof relies heavily on the assumption of "invariance to permutations" without exploring whether alternative symmetry constraints might yield other valid loss functions. Additional exploration into the necessity and sufficiency of this invariance condition could strengthen the claim.

The uniqueness result is also tied to specific choices of additive and multiplicative constants, which can limit generalization. The paper could benefit from detailing if the uniqueness result holds when different constants are applied or providing a clearer justification for these choices.  If invariance to permutations is essential to achieve the robustness and symmetry desired, the authors should provide a more detailed justification. They could demonstrate why this particular form of symmetry best aligns with robustness under label noise or show through theoretical or empirical comparisons that other symmetry constraints yield suboptimal results.

2- Over-Reliance on Assumptions in Proposition 5.2
Proposition 5.2 requires the loss function to be both non-increasing and symmetric with respect to permutations at specific points. This assumption is restrictive and could limit the practical applicability of the result, especially when the conditions on gradients may not hold in practical, noisy scenarios. It may be beneficial to relax these assumptions or provide an empirical analysis of their validity in real-world applications.

The discussion around local symmetry might also benefit from examining cases where symmetry does not hold uniformly across the input space. Providing examples or simulations where these assumptions are invalidated would help to contextualize the limitations of the theorem.

3- While the linear approximation of the cross-entropy loss around specific points is theoretically compelling, there is limited empirical evidence provided to demonstrate the robustness of this approximation in practice. Testing how this approximation affects model behavior under various noise conditions could clarify its practical utility and limitations.
The authors could empirically test the linear approximation under different noise settings. including but not limited to:
Uniform noise,
Class-conditional noise,
Asymmetric noise,
For each noise type, it would be useful to assess metrics like the model’s convergence rate, generalization on clean data, and stability over training epochs. This analysis would give a more concrete understanding of how the linear approximation affects learning dynamics in noisy settings.

4-The introduction of β-smoothness in Proposition 5.3, which bounds the size of the linear approximation's remainder, lacks a clear justification regarding how this affects overall model performance and robustness. Furthermore, the relationship between β-smoothness and generalization to unseen data remains unexplored. Further discussion on the implications of β-smoothness and empirical verification could substantiate the proposition.
Additionally, the selection of β and its sensitivity to model hyperparameters (e.g., regularization strength, learning rate) could impact the robustness claim, an area not thoroughly investigated.

5- The interpretation of the unhinged multi-class data centroid as a kernel learning component is intriguing but underdeveloped. The appendix briefly introduces this concept without exploring its potential applications in kernel-based methods or examining how the centroid relates to noise robustness.
The explanation could be made clearer by providing more context or examples of how this centroid could be leveraged in practical kernel learning setups. Additionally, the generalization to non-linear hypothesis spaces could be further clarified.
The relationship between the centroid and noise robustness should be elaborated. For instance, if the centroid can help in characterizing the loss landscape under noisy conditions, this could provide a basis for developing noise-resilient learning frameworks. An empirical investigation showing the impact of varying centroid values on robustness would also be informative.

Comparing this approach with other centroid-based noise-robust methods, such as loss decomposition and centroid estimation (Ding et al., 2022), would help in positioning the proposed method within the broader context of robust learning.

6-Proposition 4.1 introduces a decomposition method to achieve a symmetric loss function. However, the paper does not thoroughly address potential limitations or boundary cases where this decomposition might fail or yield unintended behavior, such as in highly imbalanced datasets.
While the uniqueness claim in the decomposition is theoretically interesting, the paper does not fully explore alternative decomposition approaches that might yield other symmetric loss functions. Additional comparative analysis on the benefits of this specific decomposition relative to others could enhance understanding.

7- The symmetric noise condition discussed in Appendix A, while foundational, could benefit from more discussion about its limitations. Real-world noise distributions are rarely perfectly symmetric, and a broader exploration of how deviations from symmetry impact the loss function's robustness would add practical relevance.
The appendix could also benefit from incorporating empirical data on how well this assumption holds in real-world datasets or from proposing adjustments to handle non-symmetric noise more effectively.

8- In Appendix G, the implementation details, particularly regarding Euclidean normalization and batch normalization for stability, hint at potential computational overhead. However, the appendix lacks an in-depth analysis of these computational costs, especially for large-scale datasets or real-time applications.
Additional experimentation showing the computational impact of these normalization techniques and alternative methods for improving numerical stability would be valuable.

9- In Appendix B, Proposition B.1's explanation of binary symmetric loss decomposition is somewhat unclear, especially regarding the relationship between odd and even Taylor series coefficients. More detailed explanations and illustrative examples would make this complex derivation more accessible.
Furthermore, extending this analysis to illustrate how the concept translates (or fails to translate) to the multi-class case would enhance understanding and show the limitations of applying binary assumptions to multi-class problems.

**Questions:**

1- How does the proposed symmetrization method compare with other noise-tolerant loss functions in terms of theoretical robustness and generalization?
Given the vast literature on noise-tolerant loss functions, comparing symmetrization with other loss correction approaches, such as the active-passive loss framework or transition matrix estimation, could provide insights into when and why this method might outperform or underperform. How does symmetrization fare in scenarios with non-uniform or class-conditional noise, which are common in real-world datasets?

2- What is the impact of the proposed loss functions on generalization performance under non-ideal noise conditions?
The paper’s symmetrization method assumes uniform noise distribution. However, real-world datasets often exhibit more complex noise patterns (e.g., class-conditional or asymmetric noise). Can the authors empirically or theoretically verify the method’s robustness and generalization capabilities under such more complex noise settings? How might the proposed loss functions be modified to account for non-uniform noise?

3- Could alternative constraints besides “invariance to permutations” produce comparable or even superior symmetric loss functions?
The uniqueness of the multi-class unhinged loss relies on the specific constraint of invariance to permutations, yet other forms of symmetry might yield alternative robust loss functions. For instance, can partial or relaxed symmetry conditions, such as those used in semi-supervised learning, yield alternative loss functions that retain noise robustness? Is there theoretical or empirical evidence that such alternatives might be beneficial?

4- How sensitive are SGCE and α-MAE to variations in hyperparameters across diverse model architectures and datasets?
Robustness in noisy label settings often depends on careful tuning of hyperparameters, and the success of SGCE and α-MAE could vary across different architectures (e.g., convolutional neural networks vs. transformers) or task-specific datasets. What guidelines or empirical evidence can the authors provide for tuning these parameters effectively across different architectures? Is there a general heuristic or principle to guide practitioners in tuning α for α-MAE?

5- What insights do the linear approximation and β-smoothness properties provide for real-world optimization stability and convergence?
β-smoothness in Proposition 5.3 is noted to bound the approximation’s remainder, but the implications for optimization stability, convergence speed, and computational efficiency remain unclear. Are there insights from optimization theory or empirical results that can validate these assumptions? Specifically, does β-smoothness impact the rate of convergence in practice, and does it allow for more stable training trajectories in noisy settings?

6 - How does the unhinged multi-class data centroid introduced in Appendix C compare with centroids used in clustering algorithms or kernel-based methods?
The multi-class unhinged data centroid shares conceptual similarities with centroids in clustering algorithms or kernel learning. Could a comparison be drawn here, especially in terms of interpretability and robustness to outliers? Additionally, would this centroid calculation be feasible in settings with limited computation resources, or could it be approximated?

7 - Can symmetrization be extended or adjusted to support the robust training of imbalanced datasets?
Many practical datasets contain imbalances across classes, which can exacerbate the effects of noise. Could the symmetrization method accommodate such scenarios, possibly by adjusting for class frequencies or by incorporating additional regularization terms?  How does the multi-class unhinged loss perform under varying levels of class imbalance, and are there empirical or theoretical justifications for any observed effects?

8- How does the proposed symmetrization framework align with recent advances in self-supervised or unsupervised learning under label noise?
With recent progress in self-supervised learning, particularly for handling label noise by avoiding reliance on labels entirely, it would be valuable to contextualize this symmetrization approach. Are there synergies between the proposed loss functions and self-supervised frameworks? Could the symmetrization approach be adapted or serve as a regularization term within self-supervised pretraining paradigms?

9- What are the theoretical implications of Proposition B.1’s Taylor series-based decomposition for the binary case, especially regarding generalization to non-linear classifiers?
Proposition B.1’s odd/even decomposition for the binary case through Taylor expansion is interesting but highly idealized. How does this decomposition behave in settings where non-linear classifiers (e.g., deep networks) introduce significant non-linearity? Could this decomposition help develop a more general approach for generating symmetric loss functions?

---

### Meta-Review · Area_Chair_N8GY · 2024-12-09

**Metareview:**

Noisy-label learning is an area of research that tries to learn the classifier well despite the label noise in the training datasets. This paper proposes a principled symmetrization method for designing robust loss functions for this problem setup by the decomposition of multi-class loss function into a sum of symmetric loss function and a class-insensitive term, and proposes the symmetrization of the generalized cross-entropy loss function and $\alpha$-MAE. Experiments demonstrate the competitive performance of the proposed methods.

The reviewers have noted that the proposed symmetrization method is novel and the theoretical insights add to the contribution of the paper. Reviewers also mentioned that SGCE and $\alpha$-MAE losses are also practically important contributions. Some reviewers felt the experiments are extensive and support the claims of the paper.

On the other hand, many reviewers had a hard time reading the paper, and suggested to improve writing and organization, adding clearer definitions earlier, etc. There were also concerns about the experimental setup-a reviewer pointed out that the experiments do not strictly follow the settings of previous baselines. A reviewer had novelty concerns, and mentioned that many of the properties of the multi-class unhinged loss function seem to be straightforward.

The authors provided a response to many of the concerns but the many questions/concerns of pQfd were left unaddressed. The final ratings were 65553. Based on the reviews, rebuttal, and the reasoning above, we would like to recommend rejection at this stage of the paper. However, since the paper shows good experimental results and theoretical insights, and it works on an important problem in the research community, we hope the feedback at this round is helpful to improve the organization and clarity of the novelty in the future versions of the paper.

**Additional Comments On Reviewer Discussion:**

After the authors provided the rebuttal to 4 out of the 5 reviews, there were minimal discussions during the discussion period. Two reviewers provided a reply, but two reviewers did not acknowledge that they read the rebuttal. Another reviewer did not provide a reply because the rebuttal was not posted.

---

### Decision · Program_Chairs · 2025-01-22

Reject